# Genes regulating membrane-associated E-cadherin and proliferation in *adenomatous polyposis coli* mutant colon cancer cells: High content siRNA screen

Lauren E. King[1,2], Hui-Hua Zhang[1,2], Cathryn M. Gould[3], Daniel W. Thomas[3¤], Lachlan W. Whitehead[1,2], Kaylene J. Simpson[3,4], Antony W. Burgess[1,2,5]*, Maree C. Faux[1,2]*

1 Walter and Eliza Hall Institute of Medical Research, Parkville, VIC, Australia, 2 Department of Medical Biology, University of Melbourne, Parkville, VIC, Australia, 3 Victorian Centre for Functional Genomics, Peter MacCallum Cancer Centre, Melbourne, VIC, Australia, 4 The Sir Peter MacCallum Department of Oncology, University of Melbourne, Parkville, VIC, Australia, 5 Department of Surgery, RMH, University of Melbourne, Parkville, VIC, Australia

¤ Current address: St Vincent's Institute of Medical Research, Fitzroy, VIC, Australia
* faux@wehi.edu.au (MCF); tburgess@wehi.edu.au (AWB)

**Data Availability Statement:** All relevant data are within the manuscript and its Supporting Information files.

## Abstract

Truncating mutations in the tumour suppressor gene *APC* occur frequently in colorectal cancers and result in the deregulation of Wnt signalling as well as changes in cell-cell adhesion. Using quantitative imaging based on the detection of membrane-associated E-cadherin, we undertook a protein coding genome-wide siRNA screen to identify genes that regulate cell surface E-cadherin in the APC-defective colorectal cancer cell line SW480. We identified a diverse set of regulators of E-cadherin that offer new insights into the regulation of cell-cell adhesion, junction formation and genes that regulate proliferation or survival of SW480 cells. Among the genes whose depletion promotes membrane-associated E-cadherin, we identified *ZEB1*, the microRNA200 family, and proteins such as a ubiquitin ligase UBE2E3, CDK8, sorting nexin 27 (SNX27) and the matrix metalloproteinases, MMP14 and MMP19. The screen also identified 167 proteins required for maintaining E-cadherin at cell-cell adherens junctions, including known junctional proteins, *CTNND1* and *CTNNA1*, as well as signalling enzymes, DUSP4 and MARK2, and transcription factors, TEAD3, RUNX2 and TRAM2. A better understanding of the post-translational regulation of E-cadherin provides new opportunities for restoring cell-cell adhesion in *APC*-defective cells.

## Introduction

Colorectal cancer (CRC) causes almost 862,000 deaths globally per year (World Health Organisation). Truncating mutations and subsequent loss of the *APC* (*adenomatous polyposis coli*) gene is the most common genetic event found in colon adenomas and CRC (75–90% of tumours [1]). Identifying the genes which can reverse key biological changes associated with the loss of *APC* in CRC cells has broad therapeutic implications. We have investigated the genes which regulate the levels of membrane-associated E-cadherin the CRC cell line SW480. Although APC is implicated

**Funding:** This work was supported by the Ludwig Institute for Cancer Research, the National Health and Medical Research Council (NH&MRC) Australia Program grant #487922 and the Walter and Eliza Hall Institute. The Victorian Centre for Functional Genomics (K.J.S.) is funded by the Australian Cancer Research Foundation (ACRF), the Australian Phenomics Network (APN) through funding from the Australian Government's National Collaborative Research Infrastructure Strategy (NCRIS) program and the Peter MacCallum Cancer Centre Foundation. The funders had no role in study design, data collection and analysis, decision to publish, or preparation of the manuscript.

**Competing interests:** The authors have declared that no competing interests exist.

in the control of canonical Wnt signalling [2, 3], APC is also involved in a range of other cellular processes including cell migration [4, 5], cell-cell adhesion [6, 7], mitosis [8] and differentiation [5, 9]. These processes are disrupted when *APC* is truncated.

Restoration of full-length APC in *APC* mutant CRC cells reduces β-catenin/TCF/LEF1 signalling [7, 9] and in SW480 cells the expression of full-length APC leads to functional adhesion junctions, reduced colony growth in soft agar and the SW480-APC cells no longer grow as xenografts in mice [7]. The restoration of full length APC has been accomplished *in vivo* using a transgenic shAPC/Kras$^{G12D}$/p53$^{fl/fl}$ mouse model where the expression of APC induced a sustained regression of a colorectal adenocarcinoma and re-establishment of colon crypt homeostasis [9].

The loss of functional APC reduces cell-cell adhesion [10]. β-catenin was discovered as a component of an adhesion junctions complex, where it binds to the cytoplasmic tail of E-cadherin in conjunction with α-catenin and links E-cadherin to the actin cytoskeleton [11]. There has been much debate as to whether canonical Wnt signalling and cell-cell adhesion are linked via the regulatory state of β-catenin phosphorylation [12]. E-cadherin can suppress the oncogenic potential of activated β-catenin in models of mouse colon cancer [13]. Thus E-cadherin appears to have a role in tumour suppression, as a regulator of Wnt signalling and cell-cell adhesion. In *APC* mutant and *CTNNB1* mutant adenomas and early stage CRC, E-cadherin expression at the cell membrane appears normal [14]. However, at later stages of CRC, especially in cells at the invasive front, there is a loss of E-cadherin and an increase in nuclear β-catenin [1]. A role for E-cadherin in tumorigenesis is underscored by the increased intestinal tumour burden in Apc$^{+/1638N}$/Cdh1$^{+/-}$ mice compared to mice with the Apc$^{+/1638N}$ mutation [15]. The potential of E-cadherin to inhibit the progression of CRCs [13] highlights the need to understand the regulation of E-cadherin levels, location and function in CRC cells with depleted and/or truncated APC.

Previous genome-wide studies in CRC models identified genes in the CRC cell line DLD-1 which regulate Wnt/β-catenin signalling in the context of an APC mutation [16, 17]; however, it is unclear whether the apparent effects on Wnt signalling are due to changes in the levels of cell surface E-cadherin. In this study, we performed a protein coding genome-wide screen to identify genes that regulate membrane-associated E-cadherin in *APC* mutant CRC cells. Loss of *APC* in CRC cells decreases the level of cell surface E-cadherin [18]. Using a high-throughput screen with siRNA SMARTpools, we have identified genes which suppress or stimulate the level of E-cadherin at the membrane of SW480 cells [19]. Validation and stringent gene-exclusion filters help to eliminate off-target or indirect effects [20] and resulted in the identification of 34 negative regulators and 167 positive regulators of membrane associated E-cadherin. In addition to confirming genes known to regulate the levels of membrane associated E-cadherin, we discovered a number of genes that had not previously been associated with E-cadherin regulation. This screen also revealed that miR200 family members were powerful stimulators of membrane E-cadherin. As well as regulation of E-cadherin, the screen identified anti-proliferative and pro-survival genes, however, there was no relationship between the expression of these genes and E-cadherin levels. Our imaging based screen has identified genes that can regulate membrane-associated E-cadherin in CRC cells deficient for APC and provide a new understanding of the complex mechanisms governing E-cadherin and cell-cell adhesion. The manipulation of genes that can regulate the level of E-cadherin at cell-cell junctions provides potential new targets for therapeutic intervention in CRC.

## Results

### A genome-wide imaging-based screen to identify regulators of membrane associated E-cadherin in SW480 colorectal cancer cells

To investigate genes that can regulate membrane associated E-cadherin (Ecad) in the *APC* mutant SW480 CRC cells we carried out a quantitative high-content imaging-based siRNA

screen of 18120 protein coding genes. We used siRNA knockdown of *ZEB1* and *CDH1* as controls to measure increases or decreases in membrane-associated E-cadherin levels, respectively (Fig 1). We developed an image processing algorithm to quantitate automatically the membrane associated E-cadherin (referred to as the Ecad score) (Fig 1 & S1 Fig). Untreated SW480 cells display E-cadherin in occasional patches of cells (Fig 1C); these E-cad⁺ patches disappear when the cells are exposed to siCDH1; in contrast, when SW480 cells are treated with si*ZEB1* most cells express membrane associated E-cadherin (Fig 1A and 1B). As expected, in the mock-transfected cells, there is a small proportion of SW480 cell clusters with membrane-associated E-cadherin staining (Fig 1C). When the cells are transfected with si*ZEB1*, E-cadherin membrane staining is significantly increased in almost all the cells (Fig 1A and 1D). Conversely, none of the SW480 cells treated with si*CDH1* have E-cadherin staining at the membrane (Fig 1B) and the Ecad score is 0 (Fig 1D). ZEB1 protein was reduced following treatment with siRNAs targeting ZEB1 (Fig 1E).

An Ecad score was determined for each target gene in the primary screen in which siRNA SMARTpools targeting 18120 genes were analysed (Fig 2 and S1 Table). While the majority of siRNAs did not affect membrane-associated E-cadherin, the knock down of a relatively small number of genes (453) resulted in increased Ecad scores many comparable to the average E-cad score for si*ZEB1*. The mean E-cad scores for all of the siRNA SMARTpools were plotted relative to mock controls (Fig 2A). Intra-sample normalisation was applied using a fold change to the average mock control per plate and subsequent inter-sample normalisation across plates was applied using a robust z-score normalisation [21, 22]. Conversion of the Ecad score to the robust Z score highlights the small subset of candidate genes that increase E-cadherin upon siRNA knockdown (Fig 2B). The 5.16-fold cut-off for the 231 negative regulators of E-cadherin is equivalent to an Ecad Score of 1.6 -fold greater than mock (normalised). In addition to *ZEB1*, the screen identified genes with a known association with CRC, including *ID2* [23], *CDK8* [16], *LEF1* [24] as negative regulators of E-cadherin (Fig 2C).

We detected 188 positive regulators of membrane-associated E-cadherin, including genes which have already been implicated in E-cadherin expression, binding and/or stability, such as *CTNNBIP1*, *CTNND1*, *KIF7 and DUSP4* (Fig 2C and S1 v Table). The Z-scores for these genes are similar to that of the si*CDH1* knockdown (e.g.-1.15 for *CDH1* compared to -1.12 for *CTNND1*, Fig 2C). Thus, the high-throughput screen identified genes known to regulate membrane associated cell-cell adhesion as well as a significant number of genes which regulate the level of E-cadherin at cell-cell junctions.

## micro-RNAs regulate E-cadherin mediated cell adhesion

The gene identified as having the highest robust Z-score (*UBE2E3**) from the primary screen (Fig 2C) includes effects from an off-target knockdown associated with the micro-RNA (miR)-200 family (S2A Fig), which are known to regulate and repress ZEB1 expression [25, 26]. Given that the miR-200 family has been shown to regulate ZEB1 and E-cadherin, we investigated whether individual siRNAs from the SMARTpool contain the seed sequences similar to the miR-200 family. Specifically, we used the Dharmacon Seed Sequence Analysis pipeline (Dharmacon RNAi Technologies (Horizon Discovery), unpublished tool) to identify any siRNAs that share the same seed sequence as miRNA families (miRBase version 19) (S2A Fig). We identified eleven genes with an Ecad score of >1.6 in the SMARTpool screen that contained a single duplex siRNA that shared the same seed sequence as the miR-200 family (S2B Fig). For 10 of the 11 miR200 family of genes, only 1 of the 4 duplex siRNAs demonstrated a significantly increased E-cad score (S2B Fig) and in each case that duplex contained the same seed sequence as the miR-200 family. The ubiquitin ligase *UBE2E3* was the only gene from this

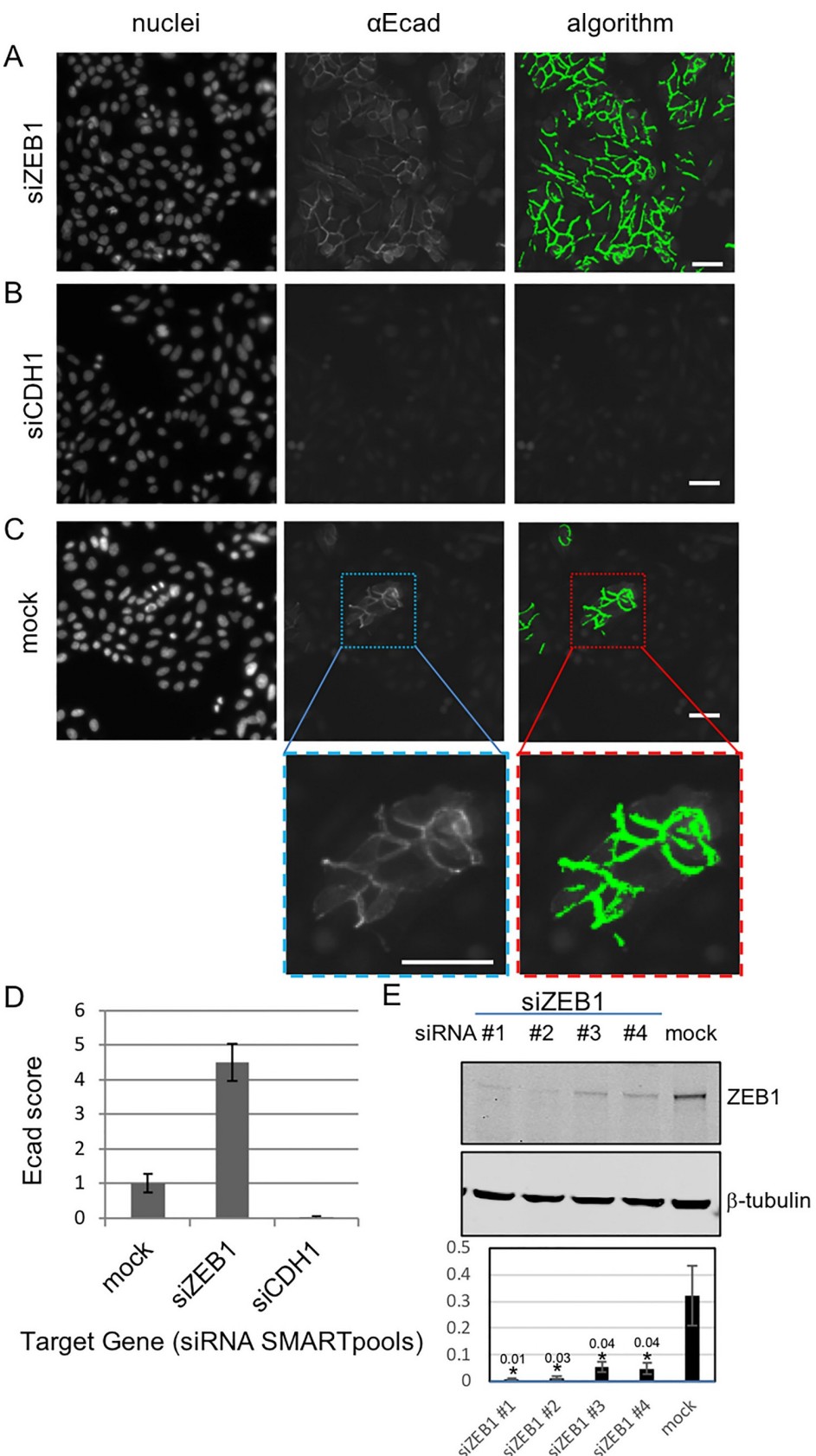

**Fig 1. Detection of membrane associated E-cadherin in SW480 cells.** Representative images from the screen showing nuclei and E-cadherin staining and the E-cadherin detection algorithm. Cells were treated with siRNA: **(A)** si*ZEB1* (increased Ecad), **(B)** si*CDH1* (decreased Ecad) and **(C)** mock controls. 72 hours later the cells were fixed, permeabilized and stained with Hoechst and αE-cadherin (HECD1). Immunofluorescent images were taken at 20x on the Cellomics ArrayScan. An E-cadherin detection algorithm was applied to the images: Scale bars; 50μm in both upper panels and insert. **(D)** The quantitation of membrane associated E-cadherin from a representative 384-well plate. Ecad score = average number of E-cadherin fibres detected per cell in a well. The Ecad score was averaged over replicate wells and normalised to the number of control (mock transfected) wells contained on every plate (Ecad score, mock n = 16 wells, si*ZEB1* and si*CDH1*, n = 6 wells, mean ± SD). **(E)** Expression of ZEB1 in SW480 cells treated with individual siRNA duplexes from the SMARTpool (siZEB1 #1–4, as indicated). ZEB1 expression is reduced with each siRNA duplex. The blot is representative of three individual experiments. Shown are cropped images, uncropped blots are included in S1 Raw images; Quantitation of ZEB1 is shown below (mean± SEM) (n = 3). Protein levels were determined using densitometry against the loading control β-tubulin *p<0.05 (exact p values are indicated); one-tailed unpaired *t*-test vs mock control.

list that when knocked down increased the E-cad score with multiple duplexes from the SMARTpool (S2B Fig). Western blot analysis confirmed that all four siRNA duplexes successfully reduced UBE2E3 levels (Fig 3A and 3B). The si*UBE2E3* duplex #1, the miR-200 mimic, elicits the largest increase in total E-cadherin with concomitant reduction in ZEB1, as expected. However, *UBE2E3* siRNA duplexes #2 and #3 also result in increased E-cadherin levels (Fig 3A and 3B) and membrane-association (Fig 3C). These duplexes did not exert corresponding changes in ZEB1 suggesting the possibility that *UBE2E3* siRNA duplexes #2 and #3 regulate membrane E-cadherin via different mechanisms of action that do not involve ZEB1. Collectively, the change in Ecad score for the siRNAs containing the seed sequence is likely due to the regulation of miR-200, however, as *UBE2E3* duplexes elicited increases in E-cadherin without altering ZEB1, UBE2E3 may regulate membrane-associated E-cadherin by a different mechanism.

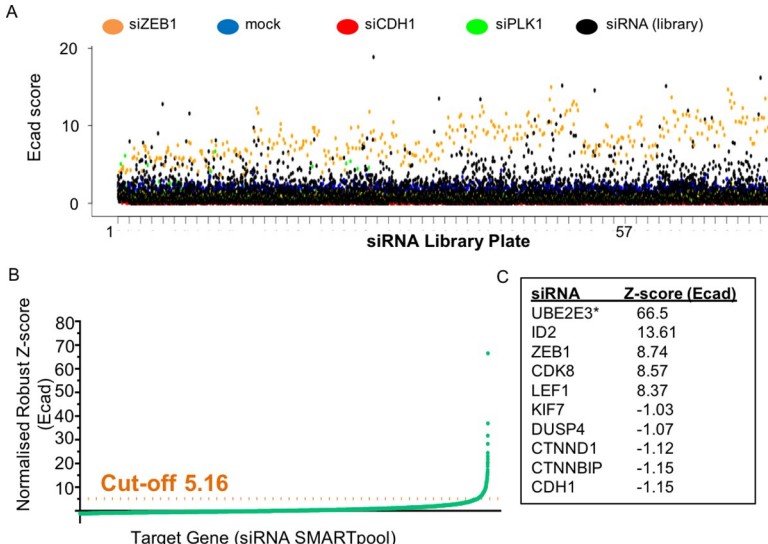

**Fig 2. A genome-wide imaging based siRNA screen identifies regulators of membrane-associated E-cadherin in SW480 colon cancer cells.** **(A)** Membrane associated Ecad scores normalised to mock transfectants for all SMARTpool siRNAs (black) transfected into SW480 cells. Controls are highlighted: mock (blue), si*ZEB1* (orange), si*CDH1* (red) & si*PLK1* (green). **(B)** Mock normalised Robust Z-score (Ecad) plot for SMARTpool siRNA screen. The cut-off for E-cadherin negative regulatory genes is indicated by the red-dotted line (Z-score>5.16); the Z- score cut-off for positive regulatory genes is <0.036. **(C)** Normalised Robust Z-score (Ecad) (Z-score Ecad) is shown for genes with a functional association with E-cadherin regulation and a gene* with potential miRNA-200 family off-target effects.

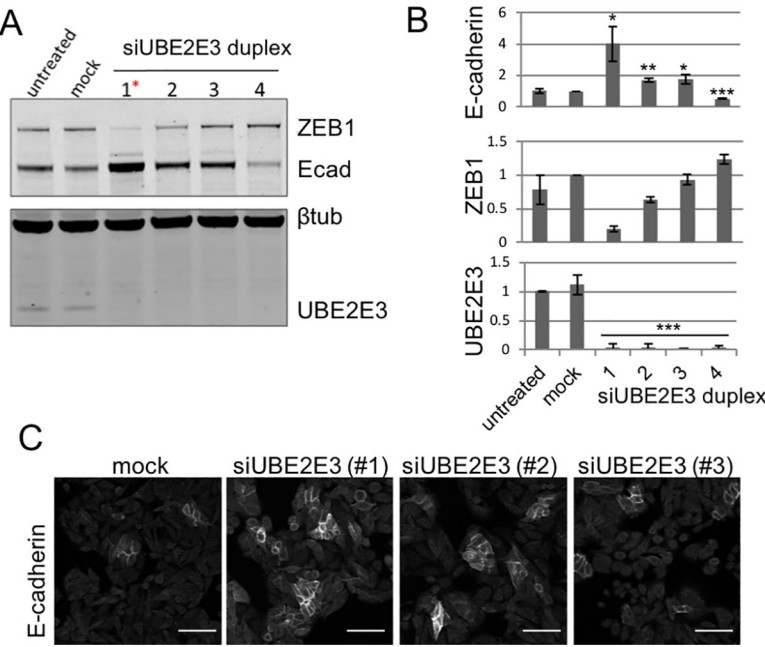

**Fig 3. Effect of siUBE2E3 on SW480 membrane-associated E-cadherin.** SW480 cells were transfected with individual siRNA duplexes from the *UBE2E3* SMARTpool for 72 hours: **(A)** Expression of ZEB1, E-cadherin and UBE2E3 were analysed by immunoblot. β-tubulin was used as a loading control. siRNA#1 (1*) shares the same 5' nucleotide sequence as miR200 family seed sequence. The blot is representative of three individual experiments. Shown are cropped images, uncropped blots are included in S1 Raw images; **(B)** Quantitation of E-cadherin, ZEB1 and UBE2E3 protein levels upon siUBE2E3 knockdown in SW480 cells. Protein levels were determined using densitometry against the loading control β-tubulin and are representative for triplicate experiments (mean± SEM) *p<0.05 (p = 0.026 and p = 0.039 for si UBE2E3 1 and 3, respectively), **p<0.005 (p = 0.00247), ***p<0.001 for E-cadherin and UBE2E3 or duplicate experiments (mean ± sd) for ZEB1 *p = 0.023, **p = 0.0035, ***p<0.001; one-tailed unpaired *t*-test vs mock control; **(C)** Immunofluorescence staining of E-cadherin in fixed SW480 cells, 72 hours post treatment with si*UBE2E3* siRNA duplexes #1, 2, 3 or mock control. Scale bar; 50μM.

## Identification of pro-survival and anti-proliferative genes

Prior to validation with the individual siRNAs from the SMART pool, the candidate gene list was refined by applying several filters to the screening analysis pipeline (S3 Fig). Firstly, cell counts were used to assess toxicity. 176 siRNAs had a clear impact on cell viability (i.e. when the cell count < 1500 cells in 25 Fields of View (FOV) [22]); these genes were excluded as specific E-cadherin regulators (S1ii Table & S3iii Fig). It should be noted that si*CDH1* did not reduce the SW480 cell counts, i.e. loss of membrane associated E-cadherin was not cytotoxic.

The high-throughput screen identified that the knock down of 176 genes reduced cell proliferation (S1 ii Table). The nuclear staining from representative fields for two pro-survival genes, *CASP8AP2* and *TUBA1B*, demonstrates the reduced cell count compared to mock control transfected cells (S4A Fig). Previous gene screens designed to detect genes essential for cell production have identified the ribosomal, mitosis and the proteasome/ubiquitin systems as important for proliferation [27, 28]. Similarly, our screen identified genes in each of these categories: 19 ribosomal genes including 14 *RPL* genes and 5 *RPS* genes; cell division and mitosis genes such as *CDK1*, *INCENP*, *KMT5a*, *PLK1* and *ZNF207*; and proteasome/ubiquitin genes such as *PSMD6*, *PSMD7*, *UBA52*, *UBB* and *UBC*. Interestingly, the earlier screen on breast cancer cell lines [28] and our screen identified that knocking down the Notch inhibitory gene *NUMB* is cytotoxic. While these 'pro-survival' genes were ruled out as specific E-cadherin regulatory genes, their expression in SW480 (mutant *APC*) cells was compared to that in SW480

+APC (restored APC) cells in order to identify genes that are differentially expressed as a result of loss of function of APC [7, 29]. We identified 7 genes (*POLR2A*, *SYNGR1*, *CST3*, *FOXD1*, *ETV3*, *OLR1*, *GRIP2*) that are important for the survival of SW480 cells. We note that *POLR2A*, *CST3*, *FOXD1* and *ETV3* also show differential expression in 4 other CRC cell lines with wild-type APC (HCT116, LIM1215, LIM1899 and RKO) [30]. As well as pro-survival, we identified 37 anti-proliferative genes whose depletion promotes SW480 cell proliferation. Genes were termed anti-proliferative based on a decreased number of FOV that were required to reach the target cell count (S1 iii Table). The average number of FOV to reach the target count of 3000 cells for the mock control wells in the primary screen was 20±3.13 (mean±SD, n = 464) (S1 vii Table). The anti-proliferative genes reached the target cell count at 2 standard deviations below the mean FOV (i.e. <15 FOV) and showed no effect on membrane associated levels of E-cadherin. SW480 cells treated with siRNAs targeting four of these genes, *ITPRIP*, *CLRN3*, *HOXC4* and *MDP1*, showed greatly increased numbers of cells/field (S4B Fig). The anti-proliferative genes included phosphatases, homeobox, cell junction and ADP-ribosylation factors. Interestingly, the depletion of magnesium-dependent phosphatase 1 (*MDP1*) has been implicated as a poor prognostic factor in gastric cancer [31]. Thus our screen identified targets that regulate proliferation in addition to pro-survival genes. Genes with an Ecad score <0.2 and >1.6-fold (positive and negative regulators, respectively) were considered as candidate E-cadherin regulators and were validated further (S3iv Fig). Genes where the expression was low (RPKM <1) [29] were removed because any changes in Ecad Score with these siRNA was considered unreliable (S3v Fig). Finally, the siRNAs containing seed sequence enrichment for the miRNA 200 family were removed (S3vi Fig).

## Novel regulators of E-cadherin

The 419 genes that appeared to regulate membrane-associated E-cadherin levels directly were re-screened using each of the four siRNAs from the SMARTpool (S3vii Fig). Genes were only considered valid hits when changes in E-cad score occurred with at least 2 siRNA [32]. The deconvolution screen identified 201 genes that regulate membrane associated E-cadherin in SW480 cells comprising 34 negative regulators of E-cadherin (si*ZEB1*-like, Ecad high genes) (S2 Table and Fig 4A) and 167 positive regulators of E-cadherin (Ecad low) (S3 Table and Fig 4A). We identified several genes that have been reported previously to be important in the regulation of cell-cell adhesion including *CTNNA1*, *CTNND1* and *ZEB1*.

Reduced E-cadherin membrane staining is evident following treatment with individual siRNA duplexes for *CTNNBIP1*, *CTNND1*, *DOCK3*, *DUSP4* and *ITGB4* (Fig 4B). Among positive regulators, we identified *CTNNBIP1* (β-catenin-TCF4 interaction inhibitor 1) a beta-catenin signalling inhibitor that also has a role in cadherin-based adhesion [33] and is upregulated in SW480+APC cells (restored APC) compared to SW480 cells [29]. The identification of transcriptional regulator *RUNX2* and its downstream target *TRAM2* as positive regulators, suggests a regulatory role for this pathway in cell-cell adhesion. Other positive regulators include *REP15*, *STAT5A*, *PI4K2A*, *ITGB4*, *MARK2, TEAD*3, *DUSP*4, *RIPK1* and *ALDH9A1*. In several cases the knockdown of these genes has been correlated with invasion and loss of E-cadherin, but this is the first time that an association between the expression of these genes and the levels of membrane associated E-cadherin has been reported. For example, *ITGB4* has been implicated in CRC progression and as a miR-21 target which represses ITGB4 leading to increased CRC cell migration [34]; *STAT5A* has been shown to promote E-cadherin and negatively regulate cell migration and invasion [35]; and *MARK2* has been linked to regulation of epithelial polarity [36]. DOCK3 (a RAC1-GEF) has been shown to regulate cell adhesion in non-small cell lung cancer cells [37]. The MAP kinase phosphatase DUSP4 has been implicated in CRC

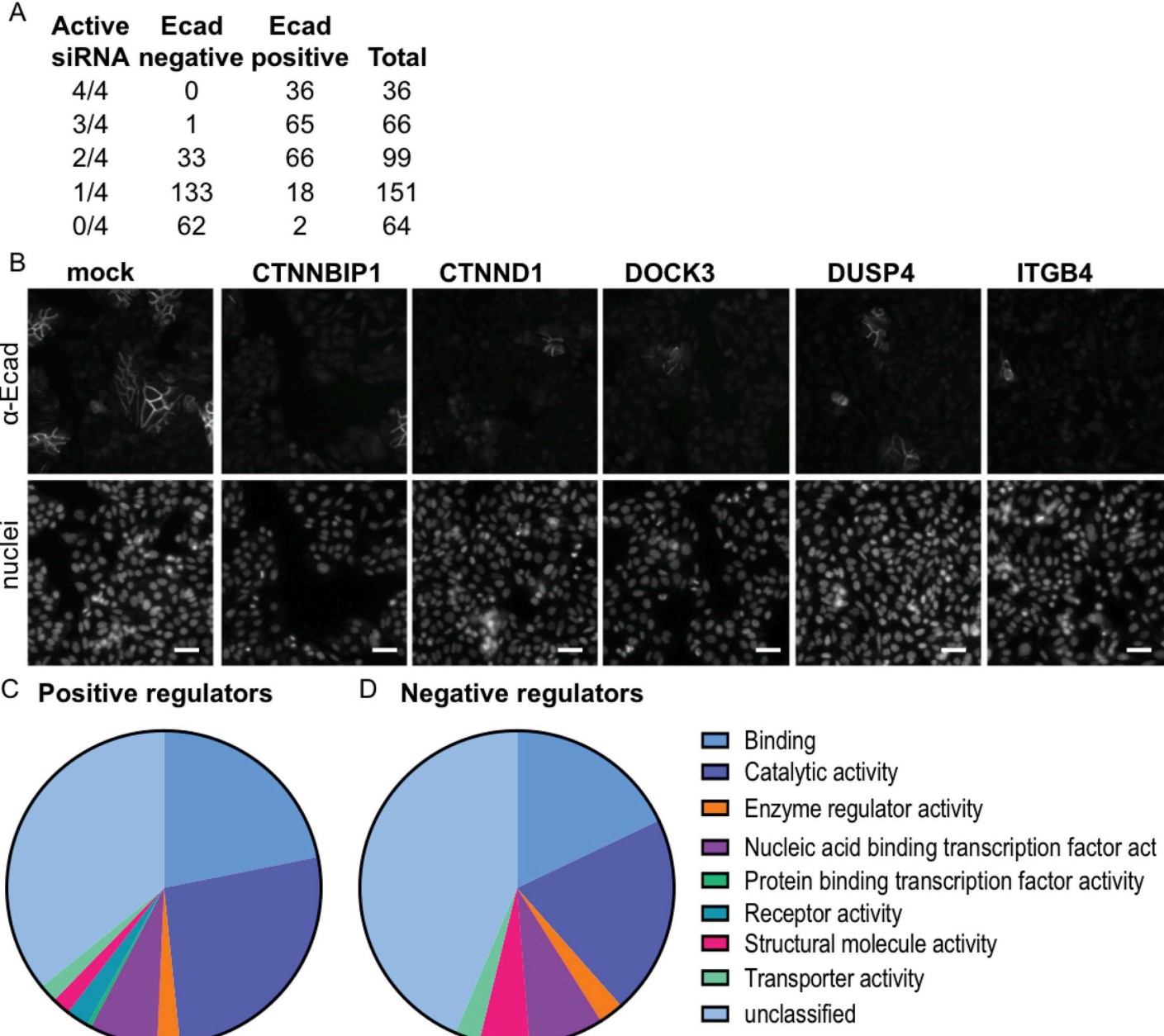

**Fig 4. siRNAs which alter SW480-membrane-associated E-cadherin. (A)** Summary of the outcome of the secondary validation screen. The number of active siRNA for E-cadherin negative and positive regulators are indicated. At least 2 out of 4 siRNA duplexes must statistically recapitulate the primary SMARTpool screen to be considered a hit. **(B)** Representative images from the deconvoluted screen of αE-cadherin and Hoechst staining, 72 hours post transfection of positive regulators with selected duplex siRNAs (25nM) for mock, *CTNNBIP1*, *CTNND1*, *DOCK3*, *DUSP4* and *ITGB4*. Scale bar; 50 μM, 20x magnification. **(C and D)** Functional categorization of candidate E-cadherin positive (C) and negative (D) regulatory genes identified from the screen (Panther Classification GO-Slim Molecular Functions, System Version 15.0).

proliferation [38] and in promoting E-cadherin expression in non-small cell lung cancer [39]. Gene ontology (GO) analysis showed that genes identified as both positive and negative regulators of E-cadherin are enriched for 'Binding' and 'Catalytic Activity' (Fig 4C and 4D; S2 and S3 Tables) and many of them function in cell-cell adhesion and endocytosis. In addition, STRING analysis which assesses and integrates protein associations [40] showed a significant

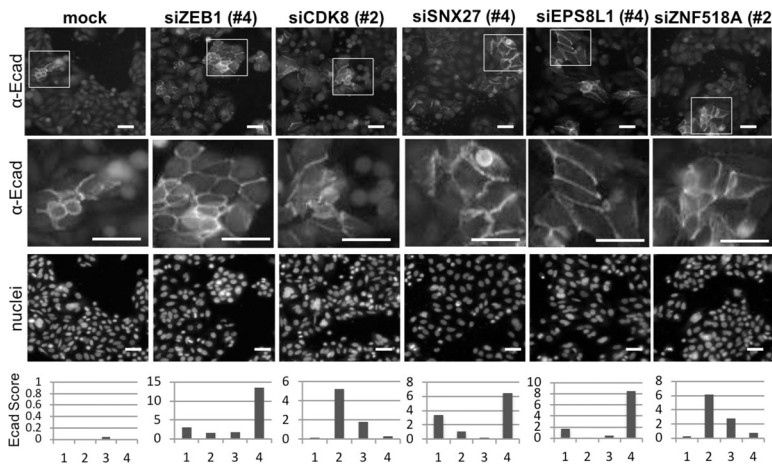

**Fig 5. SW480-membrane associated E-cadherin can increase when negative regulators are knocked down.**
Representative images from the deconvoluted screen of αE-cadherin, 72 hours post transfection of selected duplex siRNAs (25nM, duplex # indicated in brackets) for mock, *ZEB1*, *CDK8*, *SNX27*, *EPS8L1* and *ZNF518A*. Scale bar; 50 μM, 20x magnification with enlarged images for E-cadherin shown below. Hoechst staining shows cell nuclei. Mock normalised Ecad score for individual siRNA (numbered 1–4) in the deconvolution screen is shown below.

enrichment of interactions (P = 0.000124) of positive E-cadherin regulators. This included binding and catalytic activity GO terms and identified functional enrichment in cell junction organisation and signalling by receptor tyrosine kinases reactome pathways (S2 Table).

Negative regulators of E-cadherin include genes involved in a range of cellular pathways such as *CDK8*, a β-catenin regulator; *SNX27*, a member of the sorting nexin family of genes [41]; *ESP8L1*, a gene related to epidermal growth factor receptor pathway substrate 8 (EPS8) [42]; and *ZNF518A*, a zinc finger gene known to interact with histones [43] (S2 Table). E-cadherin immunostaining for *ZEB1* as well as *CDK8*, *SNX27*, *EPS8L1* and *ZNF518A* and their corresponding Ecad score for each of the four duplexes is shown in Fig 5. E-cadherin immunostaining is shown for the siRNA duplex that elicited the highest Ecad score. As distinct from the positive regulators where 4/4 and 3/4 of the siRNA duplexes reduced the levels of E-cadherin, aside from *ZEB1*, only 2/4 siRNA increased the levels of membrane associated E-cadherin for the remaining negative regulators. For *UBE2E3* 3/4 of the siRNA increased membrane-associated E-cadherin but duplex #1 contained a miR-200 seed sequence and markedly suppressed ZEB1 (see Fig 3). We investigated the potential role of sorting nexin 27 (SNX27) and matrix metalloproteinases *MMP14* and *MMP19* in more depth (S5 and S6 Figs). SNX27 facilitates recycling of endocytosed membrane-associated proteins by linking the endosome associated cargo to a retromer complex that can recycle the cargo back to the plasma membrane. Depletion of SNX27 was confirmed (S5A and S5B Fig) and resulted in increased membrane associated E-cadherin in SW480 and HCT116 cells (S5C–S5E Fig). The increased E-cadherin upon SNX27 depletion suggests a novel role for SNX27 in post-transcriptional regulation of E-cadherin. We chose the SW480+APC cells to test whether overexpression of SNX27 would reduce cell-cell adhesion in cells with functional adherens junctions. Expression of SNX27GFP, but not the functionally disrupted SNX27H114A mutant, resulted in loss of membrane associated E-cadherin in SW480+APC cells (S5F and S5G Fig). The mutation of the evolutionary conserved His114 to Ala (SNX27H114A) abrogates binding of the PDZ domain of SNX27 to PDZ binding domain containing cargo [41]. In SNX27-H114A-GFP expressing cells E-cadherin distribution is not disrupted (S5F and S5G Fig). These results show that SNX27 regulates cell-cell adhesion via an interaction with a PDZ-domain-binding

protein. We noted a reverse correlation between expression of E-cadherin and SNX27 in CRC cells (S5H Fig) and between E-cadherin and ZEB1 (S5I Fig). We found that SNX27 depletion is linked to changes in ZEB1 (S5J Fig). As depletion of ZEB1 did not change the level of SNX27 (S5K Fig) the potential regulation of E-cadherin by SNX27 is not downstream of ZEB1.

RNA sequencing revealed that *MMP14* and *MMP19* were upregulated in parental SW480 compared to SW480+APC cells (S6A Fig) [29]. *MMP14* and *MMP19* levels are also higher in HCT116 LIM1215 and LIM1899 CRC cells which contain wild-type APC [30]. *MMP14* and *MMP19* elicited Ecad scores of 1.48 and 1.56 in the primary screen, prompting further investigation. We noted that MMP14 protein levels negatively correlate with E-cadherin in CRC cell lines (S6B Fig). Depletion of *MMP14* (with siRNA#2) and *MMP19* resulted in increased E-cadherin protein but not mRNA (S6C–S6E and S6G–S6I Fig) for MMP14 and MMP19, respectively) suggesting post-transcriptional regulation of E-cadherin that results in altered membrane associated E-cadherin. Depletion of *MMP14*, but not *MMP19*, resulted in reduced ZEB1 (S6F and S6J Fig) but *ZEB1* depletion did not change the levels of either *MMP14* or *MMP19* transcript (S6D and S6H Fig). Thus MMPs 14 and 19 appear to regulate E-cadherin at a post-transcriptional level that results in membrane-associated E-cadherin.

Our imaging based screen has identified new candidate genes which regulate the level of membrane-associated E-cadherin, as well as several known positive and negative regulators of cell-cell adhesion in SW480 colorectal cancer cells (Fig 6). These results demonstrate the diversity of regulatory mechanisms which can influence the level of membrane associated E-cadherin.

## Discussion

*APC* mutation, as occurs in 80% of CRC, disrupts crypt homeostasis and results in loss of regulation of a number of biological processes: Wnt signalling (through deregulation of β-catenin),

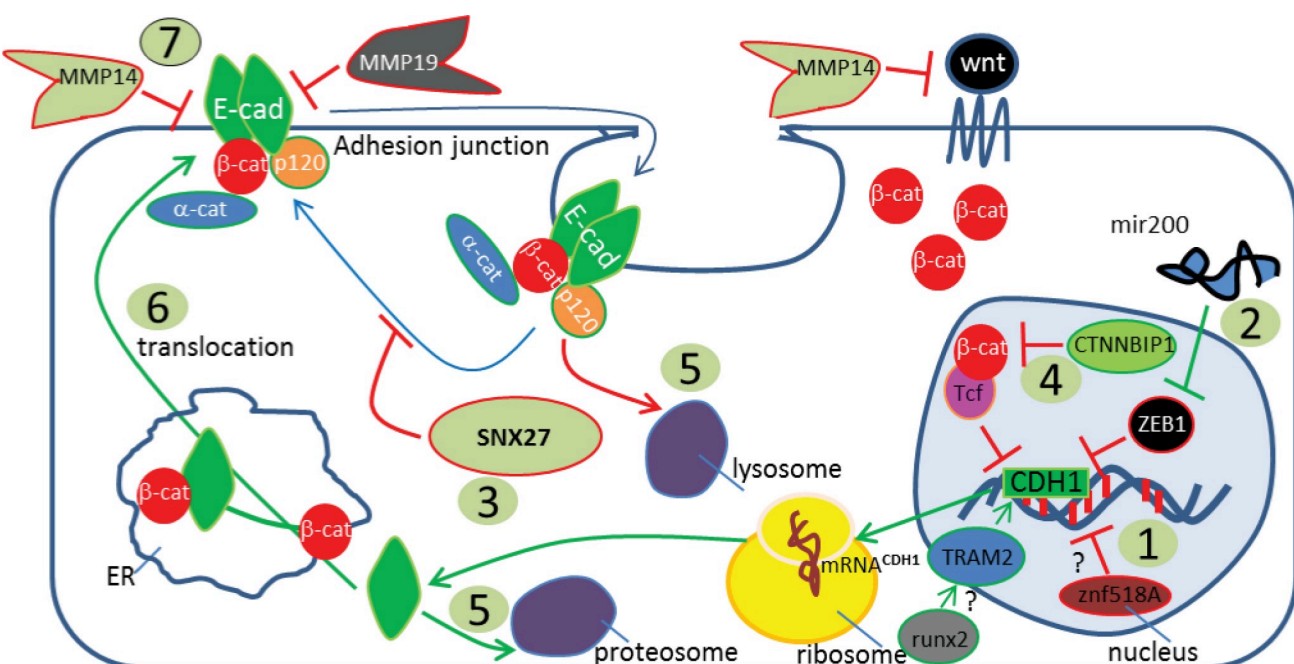

**Fig 6. Model for the regulation of junctional E-cadherin.** Schematic representation of biological processes identified in the siRNA screen for regulation of membrane-associated E-cadherin: 1. Gene transcription; 2. miR regulation (eg miR200 family); 3. Recycling of anti-adhesive or retention of pro-adhesive cargo by SNX27; 4. Inhibition of β-catenin/Tcf transcription; 5. Lysosomal or proteosomal degradation; 6. Translocation; 7. MMP14/MMP19-mediated regulation. Negative (red) and positive (green) arrows and boundaries indicate regulation of membrane-associated E-cadherin are indicated.

chromosomal stability, cell-cell adhesion and cell migration [5]. Loss of E-cadherin, the interacting partner of β-catenin at cell-cell junctions, is classically associated with an invasive cancer-cell phenotype however it has also been shown to provide a barrier to tumour development in the colon [13]. In our genome-wide imaging-based screen we have identified new processes which regulate the levels of junctional E-cadherin (Fig 6). The screen revealed 34 negative regulators (i.e. genes encoding proteins or microRNAs which decrease the level of E-cadherin) and 167 positive regulators of the levels of membrane associated E-cadherin.

In contrast to two previous E-cadherin RNAi screens [44], we were able to measure membrane associated E-cadherin directly and in an unbiased manner using a cell line that had not been modified to overexpress protein and/or viral vectors. Our screen is therefore unique in its capacity to identify genes that regulate junctional E-cadherin. The findings share some overlap with previous studies, including the identification of negative regulators Caspase 7 (*CASP7*) and Keratin 13 (*KRT13*) as well as positive regulator such as Nudix-type motif 21 (*NUDT21*) and members of the RAS oncogene family (*RAB-39 -4B* and *7A*) [44]. By incorporating our RNA-sequencing data from SW480 and SW480+APC cells in the analysis, we have also identified 7 candidates with increased expression in the tumorigenic cancer cell compared to the non-tumorigenic SW480+APC cells, that render cancer cell death upon knockdown and are therefore promising targets for colon cancer therapy.

Although we calibrated the siRNA screen to detect genes which regulate E-cadherin specifically, we also measured the effect of each siRNA on the proliferation and viability of the SW480 cells. Many of these genes have been reported as cytotoxic in other siRNA or CRISPR knockdowns [22, 45] and are presumably essential for the viability of both normal and cancer cells, e.g. the ribosomal protein, the polo kinases, DNA polymerases, centromere proteins and proteasome associated genes; however, some of the cytotoxic genes should be considered as potential targets for colon cancer therapeutics, e.g. *CDK1* [46], *CASP8AP2* [47] and tuftelin-1.

We found that among the highest scoring hits in the primary screen of the siRNA pools, individual siRNA duplexes were responsible for the increased Ecad score and these invariably contained a miR-200 family seed sequence targeting these microRNAs rather than specific genes. Regulation of E-cadherin and ZEB1 by miR-200 family members is well established [25, 26] but the results of this unbiased screen underscore the importance of the miR200 family in E-cadherin regulation. Our investigation of the miR200 seed sequences of candidate genes did enable us to identify UBE2E3 as a negative regulator that likely does not involve regulation of ZEB1, as increased E-cadherin with *UBE2E3* duplexes did not result in associated changes in ZEB1.

CDK8 has been identified as a marker of poor prognosis for patients with advanced colorectal cancer [16]. When CDK8 levels are elevated, cytoplasmic (i.e. activated) β-catenin increases. In our study reducing the levels of CDK8 leads to increased junctional E-cadherin and sequestration of β-catenin from the cytoplasm. Interestingly, BMP4 stimulates a YAP dependent increase in CDK8 [48] which would also reduce junctional E-cadherin and contribute to cancer associated EMT.

Our identification of SNX27 as a negative regulator of membrane associated E-cadherin suggest a novel function for SNX27. SNX27 has an established role in facilitating recycling of endocytosed membrane-associated proteins by linking endosome associated cargo to a retromer complex [49]. SNX27 has been implicated in regulation of ZO-2 dynamics at the cell membrane, however, in contrast to our experiments, SNX27 was shown to promote the rate of ZO-2 recovery at the membrane [50]. It is intriguing to speculate that SNX27 may have dual functions in the regulation of junctional proteins that may include recycling membrane-associated proteins or protein trafficking. Our studies point to a requirement for the PDZ domain of SNX27, through which SNX27 has been shown to interact with PDZ binding motifs of

cargo proteins [41, 49]. While the candidate cargo protein and exact mechanism for SNX27 regulation of E-cadherin is not yet clear, the SNX27 interactome which includes proteins involved in cell polarity, Wnt signalling or the shedding of cell surface receptors [51] suggest a mechanism that involves recycling proteins that are potentially 'anti-adhesive'.

Matrix metalloproteinases have been implicated in cell invasion, growth and survival. Our findings suggest that MMP14 and MMP19 also regulate membrane-associated E-cadherin. It is plausible that upregulation of MMP14 and MMP19 as a consequence of dysregulated Wnt signalling [29] promotes a tumourigenic phenotype through disruption of cell-cell adhesion, however the exact mechanism is not clear. Our data show post-transcriptional regulation of E-cadherin levels and suggest that MMP14 and MMP19 serve as modulators of membrane-associated E-cadherin, through cleavage or modification of a yet to be identified protein.

Our understanding of the role of APC defects and Wnt signalling in the development and progression of CRC is still improving. Previous studies have identified mechanisms by which β-catenin controls Wnt stimulated expression; APC depletion also changes the periphery of cells, e.g. at the cell-cell junctions [52]. These changes are characterized by decreases in E-cadherin and other cell junctional proteins. Our discovery of a small set genes which can modulate membrane associated E-cadherin levels, not only points the way to discovering mechanisms to control cell-cell junctions, but offers a new set of targets for targeting colon cancer. The most potent gene knockdowns, which increased the levels of E-cadherin, were associated with off-target effects associated with the mir200 family, so reagents which control the action of those non-coding RNAs have potential as CRC therapeutics. Similarly, drugs which target any one of the genes which inhibit the accumulation of E-cad at the membrane (e.g. UBE2E3) have the potential to inhibit oncogenesis associated with the depletion of APC.

## Materials and methods

### siRNA screening process

Cells were transfected with 40nM/well of siRNA from the Dharmacon siGENOME SMART-pool protein coding library (RefSeq v.27). The concentration of siRNA used in the deconvolution screen was 25nM/well. The screens were performed in 384 well plates. Lipofectamine 2000 (0.06µl/well) and Opti-MEM (10.94 µl/well) and siRNA were robotically dispensed (Sci-clone ALH3000, Perkin Elmer) into each well and incubated at room temperature for 15 mins. In the SMARTpool screen the following controls were used in columns 2 and 23: mock transfection, lipid and Opti-MEM, no siRNA = 16 wells, si*CDH1*/*ZEB1*/*PLK1* = 6wells. In the deconvolution secondary screen: mock = 31 wells, si*CDH1* or si*ZEB1* = 13 wells, si*PLK1* = 4 wells. Cells were then robotically dispensed (2000 cells in 25ul media, minus antibiotics, per well using a BioTek406 automated dispenser, BioTek, Vermont, USA) into each well and incubated for 72 hours. Cells were fixed (25 µl/well of 4%PFA/PBS for 10mins), permeabilised (25 µl/well of 0.2% triton-X in 0.2%BSA/PBS for 5mins) using the BioTek406. The cells were incubated with αEcad (HECD1) antibody (25 µl/well, 1:200 dilution, for 1 hour) followed by goat anti-mouse Alexa Fluor-488 secondary antibody (25 µl/well 1:500 dilution, for 1 hour in the dark) and finally incubated with Hoechst 33342 (50µM) for 10mins. Three wash steps using blocking buffer were performed between each antibody incubation. All materials were dispensed robotically. The cells were then imaged on the Cellomics ArrayScan VTi (see S1 Fig for more details).

### Antibodies

The following primary and secondary antibodies were used: anti-E-cadherin (Abcam HECD1 #ab1416 Lot# GR91484-1, 1:200), anti-E-cadherin (Cell Signalling 24E10 #3195 Lot# 04/2014,

1:1000), anti-E-cadherin (BD Transduction Laboratories #610182 Lot# 316522, 1:1000), anti-ZEB1 (Santa Cruz H-102 #sc-25388, Lot# H1513 1:1000), anti-UBE2E3 (Abcam 4B4 #ab128098 (OT14B4) Lot# GR45436-1, 1:1000), anti-SNX27 (Abcam 1C6 #ab77799 Lot# GR212908-1 and GR20549-1, 1:500), anti-MMP14 (Millipore #MAB3328 LEM-2/15.8 Lot# 2488951, 1:1000), anti-ZO1 (BD Transduction Laboratories #61096, Lot# 34962 1:200), anti-β-tubulin (Sigma Aldrich TUB2.1 T4026 Lot# 125M4884V, 1:1000), anti-occludin (Abcam #ab31721 Lot# GR115633-1, 1:200), Alexa488 goat αmouse/rabbit (#A-11001 and #A-11035 Thermo Scientific, 1:500) and Alexa546 goat αmouse/rabbit (#A-11030 and #A-11035 Thermo Scientific, 1:500).

## Cell culture conditions

SW480 cells were acquired directly from the American Type Culture Collection (ATCC CCL228). SW480 cells were cultured in RPMI supplemented with 1.08% thioglycerol, 50mg/ml hydrocortisone, 100U/ml insulin, 10% foetal calf serum (plus 1.5mg/ml G418 for the SW480+APC cells [7]. Thioglycerol is used in culture media to stimulate proliferation and we have found that this provides consistent growth conditions, but the cells do not require thioglycerol for their growth. HCT116 and Difi cells were acquired from Oliver Sieber [30]. HCT116 [53] and Difi [54] cells were cultured in DMEM supplemented with 10% foetal calf serum. All cell lines were cultured at 37 C in a 10% CO2 incubator. Cell lines were tested for mycoplasma (WEHI antibody facility) and verified to be mycoplasma free prior to starting experiments. All cell lines were authenticated by STR (short tandem repeat) profiling analysis at the Australian Genome Research Facility (AGRF) (Parkville, VIC, Australia) using the GenePrint 10 System (Promega) [30].

## siRNA transfections

$3 \times 10^5$ cells were reverse transfected with 40nM of siRNA and 5μl lipofectamine 2000 (according to manufacturer's instructions) in 6-well plate format. Twenty four hours post transfection the cells were washed once with PBS and replaced with 3mls of media. Individual siRNA for deconvolution studies (Dharmacon catalog numbers): *SNX27*: #1 D-017346-01_GUACGUAAAUUGGCACCUA; #2 D-017346-02_GGAACAACGGUUACAGUCA; #3 D-017346-03_CCAAGUAUAUCAGGCUAUC; #4 D-017346-04_GUGAAUUACUUUGCCUUAU. *MMP14*: #1 D-004145-01_GAACAAAUACUGGAAAUUC; #2 D-004145-02_GGUCUCAAAUGGCAACAUA; #3 D-004145-03_GCAAAUUCGUCUUCUUCAA; #4 D-004145-04_UCAAAUGGCAACAUAAUGA. *MMP19*: #1 D-004048-01_UGGACUACCUGUCACAAUA; #2 D-004048-03_GUGUGGCGCUACAUUAAUU;#3 D-004048-04_CUACUCGCCUCGAACACAA; #4 D-004048-18_GCGCAUCAUUGCAGCCCAU. *ZEB1* SMARTpool L-006564-01.

## Plasmid DNA transfections

$2 \times 10^6$ SW480+APC cells were plated into 60mm tissue culture dishes. The following day 2μg of pEGFP-C1 (pCTRL-GFP), pEGFP-C1-SNX27 (pSNX27-GFP) or pEGFP-C1-SNX27-H114$^{mut}$ (pSNX27-H114-GFP) were transfected using 5μl of Fugene HD into fresh media.

## RNA preparation and RT-PCR analysis

mRNA was extracted and purified using Illustra RNAspin Mini Kit (#25-0500-70 GE LifeSciences). The cDNA was prepared from mRNA using High Capacity cDNA Reverse Transcription Kit (#4368814 AB Applied Biosystems). The cDNA was then amplified in a reaction volume of 25μl using *Power*SYBR Green PCR Master Mix (#4367659 Applied Biosystems).

GAPDH was used as the house keeping gene. The samples were amplified in a 7300 Real-Time PCR system (Applied Biosystem)and the data was analysed using SDS software version 4.0 (Applied Biosystem) using the ΔΔCT method. The following primers were used: *GAPDH* FWD: CAATGACCCCTTCATTGACC, REV: TGATGACAAGCTTCCCGTTC; *CDH1* FWD: GAACGCATTGCCACATACAC, REV: ATTCGGGCTTGTTGTCATTC; *MMP14* FWD: GCAGAA GTTTTACGGCTTGC, REV: TAGCGCTTCCTTCGAACATT; *MMP19* FWD: GCTTCCTACTC CCCATGACA, REV: GCCTCGGTGATATCTTCTGG; *ZEB1* FWD: GCCAATAAGCAAACGAT TCTG, REV: TTTGGCTGGATCACTTTCAAG.

## Western blot analysis

Cells were washed once with ice-cold PBS. The cells were lysed directly using ice-cold lysis buffer (1M HEPES pH7.4, 5M NaCL, 0.5M EDTA, 10% Triton X-100, 10% Na Deoxycholate, 1x PhosStop (#04906837001 Roche) and 1x Complete EDTA-free protease inhibitor cocktail (#04693159001 Roche)) followed by 30mins incubation on ice. Lysates were centrifuged at 13,000 rpm for 30mins at 4˚C. Protein levels were standardised using a BCA protein kit (Pierce). The lysate was boiled in 2x Sample buffer (0.5M Tris-HCl pH6.8, 10% glycerol, 20% SDS (10% stock), 5% β-Mercaptoethanol, 5% Bromophenol blue (0.5% stock). Total cell lysates were analysed by SDS-PAGE (4–12% gradient gel (Thermo Fisher Scientific)), electro-trans-ferred onto nitrocellulose membrane and blocked overnight (5% skimmed milk/TBS-T) at 4˚C prior to immunoblotting with antibodies. The levels of protein were quantified using den-sitometry, normalized to β-tubulin.

## Immunofluorescence staining

Cells were fixed (4% PFA/PBS), permeabilized (0.2% Triton-X/0.2%BSA/PBS) and blocked in blocking buffer (0.2%BSA/PBS) for 1 hour at room temperature. Primary antibodies were diluted in blocking buffer and used according to manufacturer's instructions. The cells were washed 3 times in blocking buffer and secondary-fluorescent tagged antibodies were diluted 1:500 in blocking buffer and incubated for 1 hour at room temperature. The cells were washed 3 times in PBS and incubated with DAPI (#10236276001 Roche Diagnostics) or Hoechst 33342 (#B2261 Sigma-Aldrich) to stain nuclei.

## Statistical analysis

Statistical analyses were performed using an unpaired one-tailed Student's t-test, unless other-wise described. The statistical analysis for the screen is detailed in S1 and S3 Figs. Data pre-sented graphically are the means ± standard error of the mean (SEM) for three independent experiments unless otherwise stated.

## Supporting information

**S1 Fig. Screening process & image analysis.** SW480 cells were thawed from liquid nitrogen and passaged so that for each round of transfections the cells were transfected at passage 5 and at day 8 post thaw. SW480 cells were reverse transfected using lipofectamine 2000 and 40nM of SMARTpool siRNA (Dharmacon RNAi Technologies) for the primary screen, and 25nM of the individual duplex siRNA in the deconvolution screen. The siRNA library (RefSeq v.27) was accessed through, and the screen performed at, the Victorian Centre for Functional Geno-mics (VCFG), Peter MacCallum Cancer Centre. Transfections took place in optical, black walled 384-well plates. All transfections and wash steps were carried out using the Caliper Sci-clone ALH3000 and BioTek 406 liquid handling robots available at the VCFG. Cells were

washed with 40μl of PBS and replaced with 40μl of fresh media 24 hours post transfection (medium flow rate). At 72 post transfection, the cells were fixed in 4% PFA/PBS, permeabilised, immunostained with αE-cadherin (HECD1) then Alexafluor-488 αmouse antibodies and co-stained with Hoechst 34442 to mark the nuclei. The plates were then imaged on the Cellomics ArrayScan VTi HCS Reader at 20X magnification using the Cellomics Morphology V.4 Bioapplication (see S1vi Table for algorithm settings). Briefly cells were identified in channel 1 using Hoechst stain. Identification of cells allowed the algorithm to identify cell number. This count is important for cell health, proliferation and toxicity reports, and to quantify E-cadherin levels (1). The algorithm created a ring around the nuclei edge. The ring was expanded away from the cell nucleus to identify a whole cell mask for each cell. The whole cell mask is required to quantify E-cadherin (2). E-cadherin staining was identified in channel 2 using the 'fibre detection' algorithm. Briefly the algorithm parameters were set to detect long fibre-like αEcad staining (3). The Ecad score was defined as the quantity of all E-cadherin fibres from channel 2 within the modified whole cell mask from channel 1. The mean Ecad score is then quantified as the total number of fibres within the cell mask in an entire well, divided by the number of cells detected in step 1 (4).
(PDF)

**S2 Fig. siRNAs with sequence identity to the mir200 family. (A)** Gene targets with a single siRNA duplex that encodes a miR-200 family seed sequence (see S3, part vi Fig). **(B)** Dharmacon micro-RNA seed sequence analysis was carried out on the SMARTpool siRNA sequences of 454 genes. siRNAs with sequence identify to the seed sequence on the miR-200 family increased the levels of membrane-associated E-cadherin. These miRNA have a defined role in E-cadherin regulation and therefore any changes with these siRNA are likely caused by a direct effect on miRNAs rather than a specific gene.
(PDF)

**S3 Fig. Data analysis workflow.** The Dharmacon SMARTpool protein coding library comprised 18120 genes (RefSeq v.27) and was screened in 384 well format, duplicate plates per transfection (i). Raw cell count (total number of cells identified from Hoechst stain/well) and Ecad score were averaged over the duplicate plates for all controls and SMARTpool siRNAs. The total number of mock control wells were averaged per plate (16 wells per primary screen plate and 31 wells per deconvolution screen plate). The raw cell count and Ecad Scores for all SMARTpool siRNAs and the remaining control siRNAs were then normalised to the mock control (from the same plate) (ii). siRNAs were excluded from further analysis based on low cell counts (iii). siPLK1 was used as a toxicity gene control to assess and define cut-off scores for low cell count and to ensure reproducible transfection conditions each transfection. siRNA were binned into the following *Cell Viability* categories based on cell count; CV1, CV2 and *Low Count* (LC). CV1: $\geq 0.7$ -fold vs mock, CV2: $\geq 0.5 <0.7$ -fold vs mock, LC: $< 0.5$ -fold vs mock. The target cell count per well was set to 3000 and the maximum number of fields was set to 25 to be binned into CV1 category. The minimum number of cells per field was set at 14 and the maximum number of continuous sparse fields (ie fields where there are less than 14 cells) was set to 6. siRNAs in the LC category (i.e <1500 cell count in 25 FOV) were excluded from further analysis. siRNAs were removed from further analysis based on Ecad score (iv). siZEB1 and siCDH1 were used as Ecad Score positive controls to assess and define cut-off values for the high and low Ecad thresholds. siRNAs were binned into the following Ecad Score categories; High (siZEB1 like siRNA): Ecad score $1.6\geq$ -fold vs mock, NC: Ecad score $>0.2$, $<1.6$ –fold vs mock, Low (siCDH1 like siRNA): Ecad score $\leq 0.2$ –fold vs mock. siRNAs were not analysed further if they had an Ecad score in the NC category (v). RNA from SW480 cells was sequenced and analysed [1]. The siRNA targeting genes that had an RPKM of less than 1

were removed from further consideration on the premise that any changes in Ecad Score upon transfection with these siRNA may be attributed to off-target effects (v). microRNA seed sequence analysis was carried out on the SMARTpool siRNA sequences of 454 genes and compared against 3 times as many genes that had 0 Z score from the primary screen (Dharmacon RNAi Technologies unpublished program). siRNAs were removed on the basis that they had sequence identity to the seed sequence of the miRNA-200 family (vi). These miRNA have a defined role in E-cadherin regulation and therefore any changes with these siRNA are likely caused by a direct effect on miRNAs rather than a specific gene. siRNAs that passed the multiple filtering steps were then screened in the deconvolution validation screen (vii). The results from the primary screen revealed an abundant number of genes that had scored Ecad Low. The dynamic range for these genes was relatively small compared to the Ecad high genes (see S1I Table). From our own and other laboratories experience in culturing SW480 cells, we observe a higher ratio of cells with junctional E-cadherin when cells are grown to a high passage number and at increased cell density By increasing the Ecad score dynamic range between mock and siCDH1, variations in Ecad score were easier to identify between individual siRNA. We were able to increase the dynamic range by transfecting the cells at a higher passage number [2] and density (3000 cells/well) and without affecting the siCDH1 Ecad score (remains at zero). The Ecad high screen transfection was carried out under the same conditions as the primary screen. Genes were scored out of 4 individual siRNA for Ecad Score and removed from further analysis if they scored <2 active siRNAs (vii). 34 genes had a High Ecad Score (siZEB1 like genes) and 167 had a Low Ecad Score (siCDH1 like genes).
(PDF)

**S4 Fig. Pro-survival and anti-proliferative genes identified in the screen. (A)** Pro-survival genes. The number of cells/field is reduced when pro-survival genes are knocked down. Representative images from the screen of Hoechst staining 72 hours post transfection of siRNAs for mock, CASP8AP2 and TUBA1B. Scale bar; 50 μM. **(B)** Anti-proliferative genes. The number of cells/field is increased when anti-proliferative genes are knocked down. Representative images from the screen of Hoechst staining 72 hours post transfection of siRNAs for mock, *ITPRIP*, *CLRN3*, *HOXC4* and *MDP1*. Scale bar; 50 μM.
(PDF)

**S5 Fig. SNX27 is a negative regulator of membrane associated E-cadherin expression in CRC cells. (A)** Whole cell lysate immunoblot analysis siSNX27 knockdown in SW480 cells. Cells were transfected using identical siRNA oligo sequences that were used in the screen for SNX27 or ZEB1 (SMARTpool) and protein levels were assessed 72 hours later. The blot was probed with antibodies against E-cadherin, SNX27 and β-tubulin (loading control) and is representative of 4 individual experiments. **(B and C)** Quantification of SNX27 **(B)** and E-cadherin protein **(C)** levels upon siSNX27 knockdown in SW480 cells (n = 4). Protein levels were determined using densitometry against the loading control β-tubulin and displayed as the mean ± SEM For SNX27 (B) ***p<0.001 for all samples vs mock control; for E-cadherin (C) ***p<0.001 for SNX27 si #1; **p<0.01 (p = 0.006 for SNX27si #2, and p = 0.005796 for ZEB1 si), *p<0.05 (p = 0.02 and p = 0.028 for SNX27si #3 and 4, respectively), for all samples vs mock control, paired one-tailed Student's *t*-test. **(D and E)** SNX27 depletion promotes junctional E-cadherin in SW480 cells **(D)** and HCT116 cells **(E)**. E-cadherin (Ecad) (green), SNX27 (red) and nuclei (DAPI) (blue). Scale bar 50μm. **(F and G)** SNX27 regulates cell adhesion through an interaction in the SNX27PDZ domain. SNX27-eGFP expression disrupts junctional E-cadherin in SW480+APC cells **(F)** but PDZ-domain mutant, SNX27-H114A-eGFP expression does not **(G)**. Cell contacts are indicated by arrows. Junctional staining is absent in SNX27-eGFP expressing cells ** **(F)** but are intact in SNX27-H114A-eGFP

expressing cells # **(G)**. SNX27-eGFP and SNX27-H114A-eGFP (green), E-cadherin (red), nuclei (DAPI) (blue). Scale bar 50μm, left hand panels and 80μm, enlarged inset, right hand panels. **(H)** Whole cell lysate immunoblot analysis of SNX27 and E-cadherin levels in Colo320, SW480 and SW480+APC cells. β-tubulin serves as a loading control. **(I)** Immunoblot analysis of ZEB1, E-cadherin and β-tubulin in SW480 and SW480+APC cells. Shown are cropped blots, representative of three independent experiments. **(J)** ZEB1 expression upon siSNX27 knockdown in SW480 cells. Immunoblot analysis: the blot was probed with E-cadherin, ZEB1, SNX27 and β-tubulin (loading control) antibodies and ZEB1 protein levels quantified (mean ± SEM, n = 3; unpaired one-tailed Student's $t$-test, *$p < 0.05$, p values are indicated). **(K)** ZEB1 does not regulate SNX27 levels. SW480 cells were transfected with siZEB1 (SMARTpool) and protein levels were assessed 72 hours later. The blot was probed with SNX27 and β-tubulin (loading control) antibodies and are representative of two independent experiments. Quantitation is shown below (mean ± SEM, n = 2). Shown are cropped blots. Uncropped blots are included in S1 Raw images.
(PDF)

**S6 Fig. Post transcriptional regulation of E-cadherin by MMP14 and MMP19. (A)** Differential RNAseq analysis of *MMP* gene expression for SW480, SW480+APC and SW480 +control (SW480+ctrl) cells. Shown is the MEAN ± Std Dev of triplicate samples. **(B)** Whole cell lysate immunoblot analysis of MMP14 and E-cadherin in Difi, SW480 and SW480+APC cells. β-tubulin serves as a loading control. **(C)** siMMP14 knockdown (duplex#2) in SW480 cells promotes E-cadherin. E-cadherin immunoblot analysis from cells transfected with siMMP14 duplexes for 72 h. Quantification is shown in the plot below, Mean± SEM (n = 4, *p = 0.05, one-tailed unpaired Student's $t$-test vs mock control). **(D)** *MMP14* mRNA expression from SW480 cells transfected with siRNAs #1–4 or siZEB1 (SMARTpool) for 72 hours. Shown is MEAN ± SEM (n = 4), **$p < 0.01$ (p = 0.006), ***$p < 0.001$), one-tailed unpaired Student's $t$-test vs mock control. Note only duplex #2 results in depletion of *MMP14*. **(E)** *CDH1* mRNA expression from SW480 cells transfected with siMMP14 #2 or siZEB1 (SMARTpool) for 72 hours. Shown is mean ± SEM (n = 4), *$p < 0.05$ (p = 0.028), one-tailed paired Student's $t$-test vs mock control. **(F)** Whole cell lysis analysis of ZEB1 expression after knockdown of MMP14 #1–4. Quantification of ZEB1 protein levels (Mean ± SD (n = 2)) is shown below the representative blot. **(G)** siMMP19 knockdown in SW480 cells promotes E-cadherin. E-cadherin immunoblot analysis from cells transfected with siMMP19 duplexes for 72 h. Cells were harvested 72 hours post-transfection and whole cells lysates probed with antibodies against E-cadherin and β-tubulin. Quantification is shown in the plot below. Mean± SEM (n = 5) *$p < 0.05$, **$P < 0.005$ (exact p values are indicated) one-tailed unpaired Student's $t$-test vs mock control. **(H)** *MMP19* mRNA expression from SW480 cells transfected with siMMP duplexes #1–4 or siZEB1 (SMARTpool) for 72 hours. Shown is MEAN ± SEM (n = 4) *$p < 0.05$ (p = 0.048), ***$p < 0.001$ one-tailed paired Student's $t$-test vs mock control. **(I)** *CDH1* mRNA expression from SW480 cells transfected with siMMP19 duplexes #1–4 or siZEB1 (SMARTpool) for 72 hours. Shown is MEAN ± SEM (n = 4), *$p < 0.05$ (p = 0.011), one-tailed unpaired Student's $t$-test vs mock control. **(J)** Whole cell lysis analysis of ZEB1 expression after knockdown of MMP19 #1–4. For immunoblot analysis, cells were harvested 72 hours post-transfection and whole cells lysates probed with antibodies against ZEB1 and β-tubulin. Quantification of ZEB1 protein levels is shown below the representative blot. Mean ± SD (n = 4). Protein levels were determined using densitometry normalised to the loading control β-tubulin. For RNA expression, the data is normalised to *GAPDH* and shows the average of four independent experiments displayed as the Mean± SEM. Shown are cropped blots. Uncropped blots are included

in S1 Raw images.
(PDF)

**S1 Raw images.**
(PDF)

**S1 Table. Data analysis pipeline.**
(XLSX)

**S2 Table. Validated negative regulators of membrane associated E-cadherin.**
(XLSX)

**S3 Table. Validated positive regulators of membrane associated E-cadherin.**
(XLSX)

**S1 File. Supporting information reference list.**
(DOCX)

## Acknowledgments

The authors would like to thank Peter Cullen (University of Bristol, UK) for the supply of the SNX27 constructs and Oliver Sieber (Walter and Eliza Hall Institute of Medical Research, Australia) for giving us CRC cell lines.

## Author Contributions

**Conceptualization:** Lauren E. King, Kaylene J. Simpson, Antony W. Burgess, Maree C. Faux.

**Data curation:** Lauren E. King, Cathryn M. Gould, Daniel W. Thomas, Kaylene J. Simpson, Maree C. Faux.

**Formal analysis:** Lauren E. King, Cathryn M. Gould, Lachlan W. Whitehead, Kaylene J. Simpson, Maree C. Faux.

**Funding acquisition:** Antony W. Burgess.

**Investigation:** Lauren E. King, Hui-Hua Zhang, Daniel W. Thomas, Kaylene J. Simpson, Antony W. Burgess, Maree C. Faux.

**Methodology:** Lauren E. King, Hui-Hua Zhang, Daniel W. Thomas, Lachlan W. Whitehead, Kaylene J. Simpson.

**Resources:** Kaylene J. Simpson, Antony W. Burgess.

**Supervision:** Kaylene J. Simpson, Antony W. Burgess, Maree C. Faux.

**Validation:** Hui-Hua Zhang, Cathryn M. Gould, Daniel W. Thomas, Kaylene J. Simpson, Antony W. Burgess, Maree C. Faux.

**Writing – original draft:** Lauren E. King, Antony W. Burgess, Maree C. Faux.

**Writing – review & editing:** Kaylene J. Simpson, Antony W. Burgess, Maree C. Faux.

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
