## [Decision Letter · Decision Letter 0]

2 Sep 2020

PONE-D-20-23014

Genes regulating membrane-associated E-cadherin and proliferation in adenomatous polyposis coli mutant colon cancer cells: High content siRNA screen

PLOS ONE

Dear Dr. Maree C Faux,

Thank you for submitting your manuscript to PLOS ONE. After careful consideration, we feel that it has merit but does not fully meet PLOS ONE’s publication criteria as it currently stands. Therefore, we invite you to submit a revised version of the manuscript that addresses the points raised during the review process.

Please submit your revised manuscript within 60 days. If you will need more time than this to complete your revisions, please reply to this message or contact the journal office at plosone@plos.org. Please include the following items when submitting your revised manuscript:

We look forward to receiving your revised manuscript.

Kind regards,

Chunming Liu

Academic Editor

PLOS ONE

Journal Requirements:

2. Please provide additional information about the SW480, HCT116 and Difi cells used in this work, including source, history and any quality control testing procedures (authentication, characterisation, and mycoplasma testing). For more information, please see http://journals.plos.org/plosone/s/submission-guidelines#loc-cell-lines.

3. Please note that PLOS does not permit references to “data not shown.” or "not shown". Authors should provide the relevant data within the manuscript, the Supporting Information files, or in a public repository. If the data are not a core part of the research study being presented, we ask that authors remove any references to these data.

4. In the Methods section, please provide the source, product number and any lot numbers of the primary antibodies used in the Western blot and immunofluorescence analysis for your study.

5. To comply with PLOS ONE submission guidelines, in your Methods section, please provide additional information regarding your statistical analyses. For more information on PLOS ONE's expectations for statistical reporting, please see https://journals.plos.org/plosone/s/submission-guidelines.#loc-statistical-reporting.

6.Thank you for stating the following in your Competing Interests section: 

[No].

7. We note that you have indicated that data from this study are available upon request. PLOS only allows data to be available upon request if there are legal or ethical restrictions on sharing data publicly. For more information on unacceptable data access restrictions, please see http://journals.plos.org/plosone/s/data-availability#loc-unacceptable-data-access-restrictions.

8.PLOS ONE now requires that authors provide the original uncropped and unadjusted images underlying all blot or gel results reported in a submission’s figures or Supporting Information files. This policy and the journal’s other requirements for blot/gel reporting and figure preparation are described in detail at https://journals.plos.org/plosone/s/figures#loc-blot-and-gel-reporting-requirements and https://journals.plos.org/plosone/s/figures#loc-preparing-figures-from-image-files. When you submit your revised manuscript, please ensure that your figures adhere fully to these guidelines and provide the original underlying images for all blot or gel data reported in your submission. See the following link for instructions on providing the original image data: https://journals.plos.org/plosone/s/figures#loc-original-images-for-blots-and-gels.

Reviewers' comments:

Reviewer's Responses to Questions

**Comments to the Author**

1. Is the manuscript technically sound, and do the data support the conclusions?

Reviewer #1: Yes

Reviewer #2: Yes

2. Has the statistical analysis been performed appropriately and rigorously? 

Reviewer #1: I Don't Know

Reviewer #2: Yes

3. Have the authors made all data underlying the findings in their manuscript fully available?

Reviewer #1: Yes

Reviewer #2: Yes

4. Is the manuscript presented in an intelligible fashion and written in standard English?

Reviewer #1: Yes

Reviewer #2: Yes

5. Review Comments to the Author

Reviewer #1: There is a lack of consistency in the use of italics for a gene name, and capitalization needs to be used for Wnt, Notch, and others.

It would be helpful if the authors compared the findings with an APC wild type cancer cell line in the main body of the text. While they used APC expressing SW480 cells, these cells harbor an APC mutation, which may have a separate role.

The use of UBE2E3 siRNAs in Figure 3 isn't clear.

DOCK3 and DUSP4 are not discussed in the description of Figure 4.

The images used are not convincing of junctional expression of E-cadherin. Perhaps color or higher magnification should be used.

The rationale for the use of thioglycerol in the SW480 cell culture media is unclear.

The ZEB1 expression needs to be shown in Figure 1 with use of the siZEB1.

In Figure 3, it's unclear what the *, **, or *** are compared to. In addition, it is surprising that none of the values in the ZEB1 graph are significant.

In Figure 5, it would be helpful to describe how the decision is made in regards to which siRNA to show. For example, the siZEB1 (4) is shown, but according to the graph, it is the one unlike the others.

Reviewer #2: The manuscript titled “Genes regulating membrane-associated E-cadherin and proliferation in adenomatous polyposis coli mutant colon cancer cells: High content siRNA screen” identified novel candidate genes that regulate E-cadherin and may play a role in colon cancer. I recommend publishing this work after minor revisions.

1, In figure 2A, the highest Ecad score was around 20 while UBE2E3 in figure 2C showed 66.5 Ecad score. Why was UBE2E3 not included in figure 2A? What was the gene name with a highest Ecad score around 20 in figure 2A and why was it not included in figure 2C?

2, In S5 D figure, the knock down of siSNX27 was not really evident compared to mock, while the western blot results of siSNX27 in S5 A and B showed more than 50% knockdown. What was the reason?

3, At line 272, adding a brief background of the function of SNX27 and SNX27H114A mutant would help readers understand S5 F and G figures.

4, Regarding S5 F and G figures, the siRNA screen and identification of SNX27 as a negative regulator of E-cad was in SW480 cells. Why did the authors use SW480+APC cells instead of SW480 cells? If the authors also used SW480 cells to compare SNX27GFP and SNX27H114A mutant in affecting E-cad, was the results in agreement with those in SW480+APC cells?

6. PLOS authors have the option to publish the peer review history of their article (what does this mean?). If published, this will include your full peer review and any attached files.

Reviewer #1: No

Reviewer #2: No

---

## [Author Response · Author response to Decision Letter 0]

21 Sep 2020

Dear Professor Liu,

RE: PONE-D-20-23014

Genes regulating membrane-associated E-cadherin and proliferation in adenomatous polyposis coli mutant colon cancer cells: High content siRNA screen 

Thank you for your invitation to submit a revised version of our manuscript. We have addressed each of the points raised during the review process. 

The authors have declared that no competing interests exist. 

There are no restrictions to sharing the data from this study; the data are presented in the manuscript and Supporting Information files.

The original blots are included in Supporting Information in a pdf file named ‘S1_raw_images’.

Our detailed response to each of the points raised during the review process is below: 

Journal Requirements:

Response: 

The manuscript has been formatted according to the PLOS ONE’s style requirements as outlined on the style template pdf documents. 

2. Please provide additional information about the SW480, HCT116 and Difi cells used in this work, including source, history and any quality control testing procedures (authentication, characterisation, and mycoplasma testing). For more information, please see http://journals.plos.org/plosone/s/submission-guidelines#loc-cell-lines.

Response:

Additional information about the SW480, HCT and Difi cells used in this work, including source, history and quality control testing procedures is now included in the Methods Section under ‘Cell Culture Conditions’:

‘SW480 cells were acquired directly from the American Type Culture Collection (ATCC CCL228). SW480 cells were cultured in RPMI supplemented with 1.08% thioglycerol, 50mg/ml hydrocortisone, 100U/ml insulin, 10% foetal calf serum (plus 1.5mg/ml G418 for the SW480+APC cells(7). Thioglycerol is used in the culture media to stimulate proliferation. HCT116 and Difi cells were acquired from Oliver Sieber(30). HCT116(53) and Difi(54) cells were cultured in DMEM supplemented with 10% foetal calf serum. All cell lines were cultured at 37 C in a 10% CO2 incubator. Cell lines were tested for mycoplasma (WEHI antibody facility) and verified to be mycoplasma free prior to starting experiments. All cell lines were authenticated by STR (short tandem repeat) profiling analysis at the Australian Genome Research Facility (AGRF) (Parkville, VIC, Australia) using the GenePrint 10 System (Promega)(30).’

3. Please note that PLOS does not permit references to “data not shown.” or "not shown". Authors should provide the relevant data within the manuscript, the Supporting Information files, or in a public repository. If the data are not a core part of the research study being presented, we ask that authors remove any references to these data.

Response: 

Reference to “data not shown” or “not shown” have been removed.

1. We have included the data with the calculation from the mock wells in the primary screen in S1 vii Table. The text now reads:

‘The average number of FOV to reach the target count of 3000 cells for the mock control wells in the primary screen was 20±3.13 (mean±SD, n=464) (S1 vii Table).’ 

2. In the Discussion: We see reduced Wnt target gene expression upon restoration of membrane E-cadherin in MMP14 depleted cells (not shown), consistent with the idea that E-cadherin can be considered as a tumour suppressor by acting as a ‘sink’ for cytosolic �-catenin(13, 48) . However, this is not the case with MMP19.

We have not included this data in the manuscript and have removed these two sentences from the ‘Discussion’.

3. Fig 2 Figure legend: The cut-off for E-cadherin negative regulatory genes is shown (Z-score>5.16); the Z- score cut-off for positive regulatory genes is <0.036 (not shown).

This refers to the cut off for the <0.036 which is not shown on the plot.

The Figure 2B Legend now reads:

‘(B) Mock normalised Robust Z-score (Ecad) plot for SMARTpool siRNA screen. The cut-off for E-cadherin negative regulatory genes is indicated by the red-dotted line (Z-score>5.16); the Z- score cut-off for positive regulatory genes is <0.036.’

4. In the Methods section, please provide the source, product number and any lot numbers of the primary antibodies used in the Western blot and immunofluorescence analysis for your study.

Response: 

The source, product number and lots numbers of the primary antibodies used in the Western blot and immunofluorescence analysis is now included in the ‘Materials and Methods’ section: 

‘The following primary and secondary antibodies were used: anti-E-cadherin (Abcam, HECD1 #ab1416 Lot# GR91484-1, 1:200), anti-E-cadherin (Cell Signaling 24E10 #3195 Lot# 04/2014, 1:1000), anti-E-cadherin (BD Transduction Laboratories #610182 Lot# 316522, 1:1000), anti-ZEB1 (Santa Cruz H-102 #sc-25388 Lot# H1513, 1:1000), anti-UBE2E3 (Abcam 4B4 #ab128098 (OT14B4) Lot# GR45436-1, 1:1000), anti-SNX27 (Abcam 1C6 #ab77799 Lot# GR212908-1 and GR20549-1, 1:500), anti-MMP14 (Millipore #MAB3328 LEM-2/15.8 Lot# 2488951, 1:1000), anti-ZO1 (BD Transduction Laboratories #61096 Lot# 34962, 1:200), anti-β-tubulin (Sigma Aldrich TUB2.1 T4026 Lot# 125M4884V, 1:1000), anti-occludin (Abcam #ab31721 Lot# GR115633-1, 1:200), and secondary antibodies: Alexa488 goat αmouse/rabbit (Thermo Scientific #A-11001 and #A-11035, 1:500) and Alexa546 goat αmouse/rabbit (Thermo Scientific #A-11030 and #A-11035, 1:500).’

5. To comply with PLOS ONE submission guidelines, in your Methods section, please provide additional information regarding your statistical analyses. For more information on PLOS ONE's expectations for statistical reporting, please see https://journals.plos.org/plosone/s/submission-guidelines.#loc-statistical-reporting.

Response: 

A Statistical Analysis section is now included in the ‘Materials and Methods’ section:

‘Statistical Analysis

Statistical analyses were performed using an unpaired one-tailed Student’s t-test, unless otherwise described. The statistical analysis for the screen is detailed in S1 Fig and S3 Fig. Data presented graphically are the means ± standard error of the mean (SEM) for three independent experiments unless otherwise stated.’

Exact p-values are reported for all values greater than or equal to 0.001. P-values less than 0.001 are expressed as p<0.001. (eg Fig 1, 3, S5 Fig, S6 Fig).

6.Thank you for stating the following in your Competing Interests section: 

[No].

Response: 

Please see the statement in the cover letter above: “The authors have declared that no competing interests exist.”

7. We note that you have indicated that data from this study are available upon request. PLOS only allows data to be available upon request if there are legal or ethical restrictions on sharing data publicly. For more information on unacceptable data access restrictions, please see http://journals.plos.org/plosone/s/data-availability#loc-unacceptable-data-access-restrictions.

Response: 

Please see response in the cover letter above: 

‘There are no restrictions to sharing the data from this study; the data are presented in the Manuscript and Supporting Information files.’

8.PLOS ONE now requires that authors provide the original uncropped and unadjusted images underlying all blot or gel results reported in a submission’s figures or Supporting Information files. This policy and the journal’s other requirements for blot/gel reporting and figure preparation are described in detail at https://journals.plos.org/plosone/s/figures#loc-blot-and-gel-reporting-requirements and https://journals.plos.org/plosone/s/figures#loc-preparing-figures-from-image-files. When you submit your revised manuscript, please ensure that your figures adhere fully to these guidelines and provide the original underlying images for all blot or gel data reported in your submission. See the following link for instructions on providing the original image data: https://journals.plos.org/plosone/s/figures#loc-original-images-for-blots-and-gels.

Response: 

Please see response in the cover letter above: 

The original blots are included in Supporting Information in a pdf file named ‘S1_raw_images’.

Reviewers' comments:

Reviewer's Responses to Questions

Comments to the Author

1. Is the manuscript technically sound, and do the data support the conclusions?

Reviewer #1: Yes

Reviewer #2: Yes

2. Has the statistical analysis been performed appropriately and rigorously? 

Reviewer #1: I Don't Know

Reviewer #2: Yes

3. Have the authors made all data underlying the findings in their manuscript fully available?

Reviewer #1: Yes

Reviewer #2: Yes

4. Is the manuscript presented in an intelligible fashion and written in standard English?

Reviewer #1: Yes

Reviewer #2: Yes

5. Review Comments to the Author

Reviewer #1: 

There is a lack of consistency in the use of italics for a gene name, and capitalization needs to be used for Wnt, Notch, and others. 

Response: The manuscript text has been amended so that italics are consistently used for a gene name and capitalization is used for Wnt, Notch. 

It would be helpful if the authors compared the findings with an APC wild type cancer cell line in the main body of the text. While they used APC expressing SW480 cells, these cells harbor an APC mutation, which may have a separate role. 

Response: We compare SW480 cells with APC-expressing SW480 (SW480+APC) cells in the main body of the text on four occasions: 

While the SW480+APC cells express full-length APC, residual truncated APC is still expressed, as the reviewer points out. The SW480+APC cells are also a non-tumourigenic cell line and provide a model for comparison of cells with functional cell adhesion junctions. We have now compared gene expression in four other CRC cells that contain wild-type APC, but these cells are tumourigenic and contain different mutations, including other Wnt pathway genes (eg �-catenin, RNF43) which could also influence the adherens junction. 

1. We compare expression of ‘pro-survival’ genes in SW480 with the non-tumourigenic SW480+APC cells. We have now compared ‘pro-survival’ gene expression in 4 other CRC cells with wild-type APC (HCT116, LIM1215, LIM1899 and RKO). The revised text is below: 

‘While these ‘pro-survival’ genes were ruled out as specific E-cadherin regulatory genes, their expression in SW480 (mutant APC) cells was compared to that in SW480+APC (restored APC) cells in order to identify genes that are differentially expressed as a result of loss of function of APC(7, 29). We identified 7 genes (POLR2A, SYNGR1, CST3, FOXD1, ETV3, OLR1, GRIP2) that are important for the survival of SW480 cells. We note that POLR2A, CST3, FOXD1 and ETV2 also show differential expression in 4 other CRC cell lines with wild-type APC (HCT116, LIM1215, LIM1899 and RKO)(30).’ 

2. We report that expression of CTNNBIP1, a gene identified as a positive regulator, is upregulated in SW480+APC cells compared to SW480 cells. We have previously shown that SW480+APC cells demonstrate functional cell-cell adhesion junctions (Faux et al., J Cell Science 2004) and the upregulation of CTNNBIP1 which has been implicated in cadherin-based adhesion underscores the strong cell adhesion phenotype in the SW480+APC cells. CTNNBIP1 is not upregulated in HCT116, LIM1215, LIM1899 or RKO CRC cells which express wild-type APC but we suggest that this may be due to other mutations in these cells (including �-catenin).

3. In S5 Fig, expression of SNX27GFP, but not its functionally disrupted SNX27H114A mutant, results in loss of membrane associated E-cadherin in SW480+APC cells. The SW480+APC cells demonstrate robust E-cadherin at cell-cell contacts whereas other CRC cells do not show such strong junctional staining which would make interpretation of experiments with other CRC cell lines difficult. 

4. RNA sequencing revealed that MMP14 and MMP19 were upregulated in parental SW480 compared to SW480+APC cells. MMP14 and MMP19 are also upregulated in HCT116, LIM1215 and LIM1899 CRC cells. We have revised the text accordingly:

‘RNA sequencing revealed that MMP14 and MMP19 were upregulated in parental SW480 compared to SW480+APC cells (Supporting Information S6A Fig)(29). MMP14 and MMP19 levels are also higher in HCT116 LIM1215 and LIM1899 CRC cells which contain wild-type APC (30). MMP14 and MMP19 elicited Ecad scores of 1.48 and 1.56 in the primary screen, prompting further investigation.’

The use of UBE2E3 siRNAs in Figure 3 isn't clear. 

Response: The manuscript text and figure legend have been amended to clarify the use of UBE2E3 siRNAs in Figure 3 (see below). The labels on the figure have been changed to include siUBE2E3 duplex. 

‘Western blot analysis confirmed that all four siRNA duplexes successfully reduced UBE2E3 levels (Fig 3A, B). The siUBE2E3 duplex #1, the miR-200 mimic, elicits the largest increase in total E-cadherin with concomitant reduction in ZEB1, as expected. However, UBE2E3 siRNA duplexes #2 and #3 also result in increased E-cadherin levels (Fig 3A and B) and membrane-association (Fig 3C). These duplexes did not exert corresponding changes in ZEB1 suggesting the possibility that UBE2E3 siRNA duplexes #2 and #3 regulate membrane E-cadherin via different mechanisms of action that do not involve ZEB1. Collectively, the change in Ecad score for the siRNAs containing the seed sequence is likely due to the regulation of miR-200, however, as UBE2E3 duplexes elicited increases in E-cadherin without altering ZEB1, UBE2E3 may regulate membrane-associated E-cadherin by a different mechanism.’

‘Figure 3. Effect of siUBE2E3 on SW480 membrane-associated E-cadherin. SW480 cells were transfected with individual siRNA duplexes from the UBE2E3 SMARTpool for 72 hours: (A) Expression of ZEB1, E-cadherin and UBE2E3 were analysed by immunoblot. β-tubulin was used as a loading control. siRNA#1 (1*) shares the same 5’ nucleotide sequence as miR200 family seed sequence. The blot is representative of three individual experiments. Shown are cropped images, uncropped blots are included in Supporting Information S1 raw images; (B) Quantitation of E-cadherin, ZEB1 and UBE2E3 protein levels upon siUBE2E3 knockdown in SW480 cells. Protein levels were determined using densitometry against the loading control β-tubulin and are representative for triplicate experiments (mean± SEM) *p<0.05 (p=0.026 and p=0.039 for si UBE2E3 1 and 3, respectively), **p<0.005 (p=0.00247), ***p<0.001 for E-cadherin and UBE2E3 or duplicate experiments (mean ± sd) for ZEB1 *p=0.023, **p=0.0035, ***p<0.001; one-tailed unpaired t-test vs mock control; (C) Immunofluorescence staining of E-cadherin in fixed SW480 cells, 72 hours post treatment with siUBE2E3 siRNA duplexes #1, 2, 3 or mock control. Scale bar; 50µM.’

DOCK3 and DUSP4 are not discussed in the description of Figure 4. 

Response: We have amended the text to include DOCK3 and DUSP4 in the description of Figure 4. 

‘DOCK3 (a RAC1-GEF) has been shown to regulate cell adhesion in non-small cell lung cancer cells (37). The MAP kinase phosphatase DUSP4 has been implicated in CRC proliferation(38) and in promoting E-cadherin expression in non-small cell lung cancer (39).’

The images used are not convincing of junctional expression of E-cadherin. Perhaps color or higher magnification should be used. 

Response: We have included enlarged images of E-cadherin staining for SW480 cells treated with siRNAs targeting ZEB1, CDK8, SNX27, EPS8L1 and ZNF518A in Fig 5, which show increased junctional E-cadherin staining compared to mock. 

The rationale for the use of thioglycerol in the SW480 cell culture media is unclear. 

Response: Thioglycerol is used in the culture media to stimulate proliferation and has been included in the Materials and Methods ‘Cell Culture Conditions’ section. 

‘Thioglycerol is used in the culture media to stimulate proliferation.’

The ZEB1 expression needs to be shown in Figure 1 with use of the siZEB1. 

Response: We have included a new Figure, Figure 1E, showing a Western blot of Zeb1 from siZEB1 and mock transfected cells. We show a representative blot and the normalised ZEB1, quantified from 3 independent experiments. The following is included in the main text and figure legend:

‘ZEB1 protein was reduced following treatment with siRNAs targeting ZEB1 (Fig 1E).’

(E) Expression of ZEB1 in SW480 cells treated with individual siRNA duplexes from the SMARTpool (siZEB1 #1-4, as indicated). ZEB1 expression is reduced with each siRNA duplex. The blot is representative of three individual experiments. Shown are cropped images, uncropped blots are included in Supporting Information S1 raw images; Quantitation of ZEB1 is shown below (mean± SEM) (n=3). Protein levels were determined using densitometry against the loading control β-tubulin *p<0.05 (exact p values are indicated); one-tailed unpaired t-test vs mock control.

In Figure 3, it's unclear what the *, **, or *** are compared to. In addition, it is surprising that none of the values in the ZEB1 graph are significant. 

Response: The Figure 3 legend has been amended to indicate that protein levels are compared to the mock control. Statistical analysis for ZEB1 has now been included and are significant.

(B) Quantitation of E-cadherin, ZEB1 and UBE2E3 protein levels upon siUBE2E3 knockdown in SW480 cells. Protein levels were determined using densitometry against the loading control β-tubulin and are representative for triplicate experiments (mean± SEM) *p<0.05 (p=0.026 and p=0.039 for si UBE2E3 1 and 3, respectively), **p<0.005 (p=0.00247), ***p<0.001 for E-cadherin and UBE2E3 or duplicate experiments (mean ± sd) for ZEB1 *p=0.023, **p=0.0035, ***p<0.001; one-tailed unpaired t-test vs mock control;

In Figure 5, it would be helpful to describe how the decision is made in regards to which siRNA to show. For example, the siZEB1 (4) is shown, but according to the graph, it is the one unlike the others. 

Response: Representative images for E-cadherin immunostaining are shown for the siRNA duplex that elicited the highest Ecad score in order to illustrate the altered membrane E-cadherin from the screen images. The text has been amended to include the following:

 ‘E-cadherin immunostaining is shown for the siRNA duplex that elicited the highest Ecad score.’ 

Reviewer #2: The manuscript titled “Genes regulating membrane-associated E-cadherin and proliferation in adenomatous polyposis coli mutant colon cancer cells: High content siRNA screen” identified novel candidate genes that regulate E-cadherin and may play a role in colon cancer. I recommend publishing this work after minor revisions.

1, In figure 2A, the highest Ecad score was around 20 while UBE2E3 in figure 2C showed 66.5 Ecad score. Why was UBE2E3 not included in figure 2A? What was the gene name with a highest Ecad score around 20 in figure 2A and why was it not included in figure 2C? 

Response: The measurements shown in Figure 2A and Figure 2B/C are different: Figure 2A shows the Ecad score (average membrane fibre count/cell normalised to mock) and Figure 2B and C are the Normalised Robust Z-scores (Ecad). This has now been clarified in the Figure and Figure Legend. Specifically, Figure 2C now reads ‘Z-score (Ecad)’ instead of Ecad score and the Figure Legend for Figure 2C now reads: ‘Normalised Robust Z-score (Ecad) (Z-score Ecad) is shown for genes…’. The full Figure Legend is below:

‘Figure 2. A genome-wide imaging based siRNA screen identifies regulators of membrane-associated E-cadherin in SW480 colon cancer cells. (A) Membrane associated Ecad scores normalised to mock transfectants for all SMARTpool siRNAs (black) transfected into SW480 cells. Controls are highlighted: mock (blue), siZEB1 (orange), siCDH1 (red) & siPLK1 (green). (B) Mock normalised Robust Z-score (Ecad) plot for SMARTpool siRNA screen. The cut-off for E-cadherin negative regulatory genes is shown (Z-score>5.16); the Z- score cut-off for positive regulatory genes is <0.036 (not shown), (C) Normalised Robust Z-score (Ecad) (Z-score Ecad) is shown for genes with a functional association with E-cadherin regulation and a gene* with potential miRNA-200 family off-target effects.’

The gene with the highest Ecad score in Figure 2A is UBE2E3; it is included in Figure 2C with a Normalised Robust Z-score of 66.5.

2, In S5 D figure, the knock down of siSNX27 was not really evident compared to mock, while the western blot results of siSNX27 in S5 A and B showed more than 50% knockdown. What was the reason? 

Response: The signal for SNX27 in the micrographs in S5 D Figure is diminished in the siSNX27 compared to mock with corresponding increased E-cadherin. The subcellular distribution of SNX27 is concentrated in perinuclear puncta and this signal is considerably brighter in the mock treated cells. There is some background fluorescence. In the S5 D Figure, siSNX27 panel, there is a cell which shows bright fluorescent puncta, representing a non-transfected cell. The increased membrane staining for E-cadherin in the siSNX27 cells is very clear, but there is not membrane E-cadherin signal in the cell with the SNX27 perinuclear puncta. 

3, At line 272, adding a brief background of the function of SNX27 and SNX27H114A mutant would help readers understand S5 F and G figures. 

Response: The text has been amended to include the following brief description of the function of SNX27 and SNX27H114A mutant:

‘SNX27 facilitates recycling of endocytosed membrane-associated proteins by linking the endosome associated cargo to a retromer complex that can recycle the cargo back to the plasma membrane. Depletion of SNX27 was confirmed (S5A, B Fig) and resulted in increased membrane associated E-cadherin in SW480 and HCT116 cells (S5C, D, E Fig). The increased E-cadherin upon SNX27 depletion suggests a novel role for SNX27 in post-transcriptional regulation of E-cadherin. Expression of SNX27GFP, but not the functionally disrupted SNX27H114A mutant, resulted in loss of membrane associated E-cadherin in SW480+APC cells (S5F, G Fig). The mutation of the evolutionary conserved His114 to Ala (SNX27H114A) abrogates binding of the PDZ domain of SNX27 to PDZ binding domain containing cargo(41). In SNX27-H114A-GFP expressing cells E-cadherin distribution is not disrupted (S5F, G Fig). These results show that SNX27 regulates cell-cell adhesion via an interaction with a PDZ-domain-binding protein.’

4, Regarding S5 F and G figures, the siRNA screen and identification of SNX27 as a negative regulator of E-cad was in SW480 cells. Why did the authors use SW480+APC cells instead of SW480 cells? If the authors also used SW480 cells to compare SNX27GFP and SNX27H114A mutant in affecting E-cad, was the results in agreement with those in SW480+APC cells? 

Response: We chose the SW480+APC cells because these cells produce functional cell-cell adhesion contacts, with E-cadherin membrane staining. In contrast, SW480 cells show minimal E-cadherin membrane staining and further loss of junctional membrane E-cadherin is difficult to detect. The result in SW480 cells is in agreement with SW480+APC cells. The following sentence has been included in the text:

‘We chose the SW480+APC cells to test whether overexpression of SNX27 would reduce cell-cell adhesion in cells with functional adherens junctions.’

We trust that our manuscript is now suitable for publication in PLOS ONE.

Yours sincerely,

Maree Faux 

The Walter and Eliza Hall Institute of Medical Research,

1G Royal Parade, Parkville, VICTORIA 3051

Australia

Email: faux@wehi.edu.au

---

## [Decision Letter · Decision Letter 1]

28 Sep 2020

PONE-D-20-23014R1

Genes regulating membrane-associated E-cadherin and proliferation in adenomatous polyposis coli mutant colon cancer cells: High content siRNA screen

PLOS ONE

Dear Dr. Maree C Faux,

Thank you for submitting your revised manuscript to PLOS ONE. The manuscript was carefully reviewed by the original reviewers.  Although the reviewers believe that the revised manuscript was significantly improved, they still have minor concerns.  Therefore, we invite you to submit a revised version of the manuscript that addresses the points raised during the review process.

Please submit your revised manuscript within two months. If you will need more time than this to complete your revisions, please reply to this message or contact the journal office at plosone@plos.org. Please include the following items when submitting your revised manuscript:

We look forward to receiving your revised manuscript.

Kind regards,

Chunming Liu

Academic Editor

PLOS ONE

Reviewers' comments:

Reviewer's Responses to Questions

**Comments to the Author**

1. If the authors have adequately addressed your comments raised in a previous round of review and you feel that this manuscript is now acceptable for publication, you may indicate that here to bypass the “Comments to the Author” section, enter your conflict of interest statement in the “Confidential to Editor” section, and submit your "Accept" recommendation.

Reviewer #1: All comments have been addressed

Reviewer #2: All comments have been addressed

2. Is the manuscript technically sound, and do the data support the conclusions?

Reviewer #1: (No Response)

Reviewer #2: Yes

3. Has the statistical analysis been performed appropriately and rigorously? 

Reviewer #1: (No Response)

Reviewer #2: Yes

4. Have the authors made all data underlying the findings in their manuscript fully available?

Reviewer #1: (No Response)

Reviewer #2: Yes

5. Is the manuscript presented in an intelligible fashion and written in standard English?

Reviewer #1: (No Response)

Reviewer #2: Yes

6. Review Comments to the Author

Reviewer #1: While all comments have been addressed, there is still concern about the use of thioglycerol in the cell culture media for the SW480 cells. The authors state that it is needed for proliferation. While thio is used in some primary cell culture media, it should not be required for culturing the highly aggressive SW-480 cells. The concern here is that there is something inherently wrong with these cells that require thioglycerol to grow, which may also influence the results described in the manuscript.

Reviewer #2: (No Response)

7. PLOS authors have the option to publish the peer review history of their article (what does this mean?). If published, this will include your full peer review and any attached files.

Reviewer #1: No

Reviewer #2: No

---

## [Author Response · Author response to Decision Letter 1]

29 Sep 2020

Thank you for your invitation to submit a revised version of our manuscript. We are pleased to see that both reviewers agree that ‘All comments have been addressed’ and that all other requirements have been resolved. We have addressed the remaining concern raised by Reviewer 1 in our detailed response below:

6. Review Comments to the Author

Reviewer #1: While all comments have been addressed, there is still concern about the use of thioglycerol in the cell culture media for the SW480 cells. The authors state that it is needed for proliferation. While thio is used in some primary cell culture media, it should not be required for culturing the highly aggressive SW-480 cells. The concern here is that there is something inherently wrong with these cells that require thioglycerol to grow, which may also influence the results described in the manuscript.

Response: We have routinely cultured colon cancer cell lines in RPMI supplemented with 10%FCS, insulin, thioglycerol and hydrocortisone. We have found that this provides consistent growth conditions for a range of cell lines and have used this media for culturing the SW480 cells in this study. We understand the reviewer's concern for a robust cell line such as SW480 cells, but we have also grown the SW480 cells in other media such as DME plus 10% FCS and have not observed a difference in growth rate. Furthermore, the various controls that we used in the experiments in the manuscript do not show adverse effects on proliferation or other growth characteristics. We do not believe that there is something inherently wrong with these cells that require thioglycerol to grow. 

We have included the following in the revised manuscript:

‘Thioglycerol is used in culture media to stimulate proliferation and we have found that this provides consistent growth conditions, but the cells do not require thioglycerol for their growth.’

---

## [Decision Letter · Decision Letter 2]

2 Oct 2020

Genes regulating membrane-associated E-cadherin and proliferation in adenomatous polyposis coli mutant colon cancer cells: High content siRNA screen

PONE-D-20-23014R2

Dear Dr. Maree C Faux,

We’re pleased to inform you that your manuscript has been judged scientifically suitable for publication and will be formally accepted for publication once it meets all outstanding technical requirements.

Kind regards,

Chunming Liu

Academic Editor

PLOS ONE

Additional Editor Comments (optional):

Reviewers' comments:

Reviewer's Responses to Questions

**Comments to the Author**

1. If the authors have adequately addressed your comments raised in a previous round of review and you feel that this manuscript is now acceptable for publication, you may indicate that here to bypass the “Comments to the Author” section, enter your conflict of interest statement in the “Confidential to Editor” section, and submit your "Accept" recommendation.

Reviewer #1: All comments have been addressed

2. Is the manuscript technically sound, and do the data support the conclusions?

Reviewer #1: (No Response)

3. Has the statistical analysis been performed appropriately and rigorously? 

Reviewer #1: (No Response)

4. Have the authors made all data underlying the findings in their manuscript fully available?

Reviewer #1: (No Response)

5. Is the manuscript presented in an intelligible fashion and written in standard English?

Reviewer #1: (No Response)

6. Review Comments to the Author

Reviewer #1: (No Response)

7. PLOS authors have the option to publish the peer review history of their article (what does this mean?). If published, this will include your full peer review and any attached files.

Reviewer #1: No

---

## [Editor Report · Acceptance letter]

7 Oct 2020

PONE-D-20-23014R2 

Genes regulating membrane-associated E-cadherin and proliferation in *adenomatous polyposis coli* mutant colon cancer cells: High content siRNA screen 

Dear Dr. Faux:

I'm pleased to inform you that your manuscript has been deemed suitable for publication in PLOS ONE. Congratulations! Your manuscript is now with our production department. 

Kind regards, 

on behalf of

Dr. Chunming Liu 

Academic Editor

PLOS ONE